# Query Complexity of Clustering with Side Information

**Arya Mazumdar and Barna Saha**
College of Information and Computer Sciences
University of Massachusetts Amherst
Amherst, MA 01003
{arya,barna}@cs.umass.edu

## Abstract

Suppose, we are given a set of $n$ elements to be clustered into $k$ (unknown) clusters, and an oracle/expert labeler that can interactively answer pair-wise queries of the form, "do two elements $u$ and $v$ belong to the same cluster?". The goal is to recover the optimum clustering by asking the minimum number of queries. In this paper, we provide a rigorous theoretical study of this basic problem of query complexity of interactive clustering, and give strong information theoretic lower bounds, as well as nearly matching upper bounds. Most clustering problems come with a similarity matrix, which is used by an automated process to cluster similar points together. However, obtaining an ideal similarity function is extremely challenging due to ambiguity in data representation, poor data quality etc., and this is one of the primary reasons that makes clustering hard. To improve accuracy of clustering, a fruitful approach in recent years has been to ask a domain expert or crowd to obtain labeled data interactively. Many heuristics have been proposed, and all of these use a similarity function to come up with a querying strategy. Even so, there is a lack systematic theoretical study. Our main contribution in this paper is to show the dramatic power of side information aka similarity matrix on reducing the query complexity of clustering. A similarity matrix represents noisy pair-wise relationships such as one computed by some function on attributes of the elements. A natural noisy model is where similarity values are drawn independently from some arbitrary probability distribution $f_+$ when the underlying pair of elements belong to the same cluster, and from some $f_-$ otherwise. We show that given such a similarity matrix, the query complexity reduces drastically from $\Theta(nk)$ (no similarity matrix) to $O(\frac{k^2 \log n}{\mathcal{H}^2(f_+\|f_-)})$ where $\mathcal{H}^2$ denotes the squared Hellinger divergence. Moreover, this is also information-theoretic optimal within an $O(\log n)$ factor. Our algorithms are all efficient, and parameter free, i.e., they work without any knowledge of $k, f_+$ and $f_-$, and only depend logarithmically with $n$. Our lower bounds could be of independent interest, and provide a general framework for proving lower bounds for classification problems in the interactive setting. Along the way, our work also reveals intriguing connection to popular community detection models such as the *stochastic block model* and opens up many avenues for interesting future research.

## 1 Introduction

Clustering is one of the most fundamental and popular methods for data classification. In this paper we provide a rigorous theoretical study of *clustering with the help of an oracle*, a model that saw a recent surge of popular heuristic algorithms.

Suppose we are given a set of $n$ points, that need to be clustered into $k$ clusters where $k$ is unknown to us. Suppose there is an oracle that either knows the true underlying clustering or can compute the best clustering under some optimization constraints. We are allowed to query the oracle whether any two points belong to the same cluster or not. How many such queries are needed to be asked at minimum to perform the clustering exactly? The motivation to this problem lies at the heart of modern machine learning applications where the goal is to facilitate more accurate learning from less data by interactively asking for labeled data, e.g., active learning and crowdsourcing. Specifically, automated clustering algorithms that rely just on a similarity matrix often return inaccurate results. Whereas, obtaining few labeled data adaptively can help in significantly improving its accuracy. Coupled with this observation, clustering with an oracle has generated tremendous interest in the last few years with increasing number of heuristics developed for this purpose [22, 40, 13, 42, 43, 18, 39, 12, 21, 29]. The number of queries is a natural measure of "efficiency" here, as it directly relates to the amount of labeled data or the cost of using crowd workers –however, theoretical guarantees on query complexity is lacking in the literature.

On the theoretical side, query complexity or the decision tree complexity is a classical model of computation that has been extensively studied for different problems [16, 4, 8]. For the clustering problem, one can obtain an upper bound of $O(nk)$ on the query complexity easily and it is achievable even when $k$ is unknown [40, 13]: to cluster an element at any stage of the algorithm, ask one query per existing cluster with this element (this is sufficient due to transitivity), and start a new cluster if all queries are negative. It turns out that $\Omega(nk)$ is also a lower bound, even for randomized algorithms (see, e.g., [13]). In contrast, the heuristics developed in practice often ask significantly less queries than $nk$. What could be a possible reason for this deviation between the theory and practice?

Before delving into this question, let us look at a motivating application that drives this work.

**A Motivating Application: Entity Resolution.** Entity resolution (ER, also known as record linkage) is a fundamental problem in data mining and has been studied since 1969 [17]. The goal of ER is to identify and link/group different manifestations of the same real world object, e.g., different ways of addressing (names, email address, Facebook accounts) the same person, Web pages with different descriptions of the same business, different photos of the same object etc. (see the excellent survey by Getoor and Machanavajjhala [20]). However, lack of an ideal similarity function to compare objects makes ER an extremely challenging task. For example, DBLP, the popular computer science bibliography dataset is filled with ER errors [30]. It is common for DBLP to merge publication records of different persons if they share similar attributes (e.g. same name), or split the publication record of a single person due to slight difference in representation (e.g. Marcus Weldon vs Marcus K. Weldon). In recent years, a popular trend to improve ER accuracy has been to incorporate human wisdom. The works of [42, 43, 40] (and many subsequent works) use a computer-generated similarity matrix to come up with a collection of pair-wise questions that are asked interactively to a crowd. The goal is to minimize the number of queries to the crowd while maximizing the accuracy. This is analogous to our interactive clustering framework. But intriguingly, as shown by extensive experiments on various real datasets, these heuristics use far less queries than $nk$ [42, 43, 40]–barring the $\Omega(nk)$ theoretical lower bound. On a close scrutiny, we find that all of these heuristics use some computer generated similarity matrix to guide in selecting the queries. Could these similarity matrices, aka side information, be the reason behind the deviation and significant reduction in query complexity?

Let us call this *clustering using side information*, where the clustering algorithm has access to a similarity matrix. This can be generated directly from the raw data (e.g., by applying Jaccard similarity on the attributes), or using a crude classifier which is trained on a very small set of labelled samples. Let us assume the following generative model of side information: a noisy weighted upper-triangular similarity matrix $W = \{w_{i,j}\}$, $1 \le i < j \le n$, where $w_{i,j}$ is drawn from a probability distribution $f_+$ if $i, j, i \ne j$, belong to the same cluster, and else from $f_-$. However, the algorithm designer is given only the similarity matrix without any information on $f_+$ and $f_-$. In this work, one of our major contributions is to show the separation in query complexity of clustering with and without such side information. Indeed the recent works of [18, 33] analyze popular heuristic algorithms of [40, 43] where the probability distributions are obtained from real datasets which show that these heuristics are significantly suboptimal even for very simple distributions. To the best of our knowledge, before this work, there existed no algorithm that works for arbitrary unknown distributions $f_+$ and $f_-$ with near-optimal performances. We develop a generic framework for proving information theoretic lower bounds for interactive clustering using side information, and design efficient algorithms for arbitrary

$f_+$ and $f_-$ that nearly match the lower bound. Moreover, our algorithms are parameter free, that is they work without any knowledge of $f_+$, $f_-$ or $k$.

**Connection to popular community detection models.** The model of side information considered in this paper is a direct and significant generalization of the *planted partition model*, also known as the stochastic block model (SBM) [28, 15, 14, 2, 1, 25, 24, 11, 36]. The stochastic block model is an extremely well-studied model of random graphs which is used for modeling communities in real world, and is a special case of a similarity matrix we consider. In SBM, two vertices within the same community share an edge with probability $p$, and two vertices in different communities share an edge with probability $q$, that is $f_+$ is Bernoulli($p$) and $f_-$ is Bernoulli($q$). It is often assumed that $k$, the number of communities, is a constant (e.g. $k = 2$ is known as the *planted bisection model* and is studied extensively [1, 36, 15] or a slowly growing function of $n$ (e.g. $k = o(\log n)$). The points are assigned to clusters according to a probability distribution indicating the relative sizes of the clusters. In contrast, not only in our model $f_+$ and $f_-$ can be arbitrary probability mass functions (pmfs), we do not have to make any assumption on $k$ or the cluster size distribution, and can allow for any partitioning of the set of elements (i.e., adversarial setting). Moreover, $f_+$ and $f_-$ are unknown. For SBM, parameter free algorithms are known relatively recently for constant number of linear sized clusters [3, 24].

There are extensive literature that characterize the threshold phenomenon in SBM in terms of $p$ and $q$ for exact and approximate recovery of clusters when relative cluster sizes are known and nearly balanced (e.g., see [2] and therein for many references). For $k = 2$ and equal sized clusters, sharp thresholds are derived in [1, 36] for a specific sparse region of $p$ and $q$ [1]. In a more general setting, the vertices in the $i$th and the $j$th communities are connected with probability $q_{ij}$ and threshold results for the sparse region has been derived in [2] - our model can be allowed to have this as a special case when we have pmfs $f_{i,j}$s denoting the distributions of the corresponding random variables. If an oracle gives us some of the pairwise binary relations between elements (whether they belong to the same cluster or not), the threshold of SBM must also change. But by what amount? This connection to SBM could be of independent interest to study query complexity of interactive clustering with side information, and our work opens up many possibilities for future direction.

Developing lower bounds in the interactive setting appears to be significantly challenging, as algorithms may choose to get any deterministic information adaptively by querying, and standard lower bounding techniques based on Fano-type inequalities [9, 31] do not apply. One of our major contributions in this paper is to provide a general framework for proving information-theoretic lower bound for interactive clustering algorithms which holds even for randomized algorithms, and even with the full knowledge of $f_+$, $f_-$ and $k$. In contrast, our algorithms are computationally efficient and are parameter free (works without knowing $f_+$, $f_-$ and $k$). The technique that we introduce for our upper bounds could be useful for designing further parameter free algorithms which are extremely important in practice.

**Other Related works.** The interactive framework of clustering model has been studied before where the oracle is given the entire clustering and the oracle can answer whether a cluster needs to be split or two clusters must be merged [7, 6]. Here we contain our attention to pair-wise queries, as in all practical applications that motivate this work [42, 43, 22, 40]. In most cases, an expert human or crowd serves as an oracle. Due to the scale of the data, it is often not possible for such an oracle to answer queries on large number of input data. Only recently, some heuristic algorithms with $k$-wise queries for small values of $k$ but $k > 2$ have been proposed in [39], and a non-interactive algorithm that selects random triangle queries have been analyzed in [41]. Also recently, the stochastic block model with active label-queries have been studied in [19]. Perhaps conceptually closest to us is a recent work by [5] where they consider pair-wise queries for clustering. However, their setting is very different. They consider the specific NP-hard $k$-means objective with distance matrix which must be a metric and must satisfy a deterministic separation property. Their lower bounds are computational and not information theoretic; moreover their algorithm must know the parameters. There exists a significant gap between their lower and upper bounds:$\sim \log k$ vs $k^2$, and it would be interesting if our techniques can be applied to improve this.

Here we have assumed the oracle always returns the correct answer. To deal with the possibility that the crowdsourced oracle may give wrong answers, there are simple majority voting mechanisms or more complicated techniques [39, 12, 21, 29, 10, 41] to handle such errors. Our main objective is to

study the power of side information, and we do not consider the more complex scenarios of handling erroneous oracle answers. The related problem of clustering with noisy queries is studied by us in a companion work [34]. Most of the results of the two papers are available online in a more extensive version [32].

**Contributions.** Formally the problem we study in this paper can be described as follows.

**Problem 1** (Query-Cluster with an Oracle). *Consider a set of elements $V \equiv [n]$ with $k$ latent clusters $V_i$, $i = 1, \dots, k$, where $k$ is unknown. There is an oracle $\mathcal{O} : V \times V \to \{\pm 1\}$, that when queried with a pair of elements $u, v \in V \times V$, returns $+1$ iff $u$ and $v$ belong to the same cluster, and $-1$ iff $u$ and $v$ belong to different clusters. The queries $Q \subseteq V \times V$ can be done adaptively. Consider the side information $W = \{w_{u,v} : 1 \le u < v \le n\}$, where the $(u,v)$th entry of $W$, $w_{u,v}$ is a random variable drawn from a discrete probability distribution $f_+$ if $u, v$ belong to the same cluster, and is drawn from a discrete[2] probability distribution $f_-$[3] if $u, v$ belong to different clusters. The parameters $k$, $f_+$ and $f_-$ are unknown. Given $V$ and $W$, find $Q \subseteq V \times V$ such that $|Q|$ is minimum, and from the oracle answers and $W$ it is possible to recover $V_i$, $i = 1, 2, ..., k$.*

Without side information, as noted earlier, it is easy to see an algorithm with query complexity $O(nk)$ for Query-Cluster. When no side information is available, it is also not difficult to have a lower bound of $\Omega(nk)$ on the query complexity. Our main contributions are to develop strong information theoretic lower bounds as well as nearly matching upper bounds when side information is available, and characterize the effect of side information on query complexity precisely.

**Upper Bound (Algorithms).** We show that with side information $W$, a drastic reduction in query complexity of clustering is possible, even with unknown parameters $f_+$, $f_-$, and $k$. We propose a Monte Carlo randomized algorithm that reduces the number of queries from $O(nk)$ to $O(\frac{k^2 \log n}{\mathcal{H}^2(f_+ \| f_-)})$, where $\mathcal{H}(f \| g)$ is the Hellinger divergence between the probability distributions $f$, and $g$, and recovers the clusters accurately with high probability (with success probability $1 - \frac{1}{n}$) without knowing $f_+$, $f_-$ or $k$ (see, Theorem 1). Depending on the value of $k$, this could be highly sublinear in $n$. Note that the squared Hellinger divergence between two pmfs $f$ and $g$ is defined to be,

$$\mathcal{H}^2(f \| g) = \frac{1}{2} \sum_i \left( \sqrt{f(i)} - \sqrt{g(i)} \right)^2.$$

We also develop a Las Vegas algorithm, that is one which recovers the clusters with probability 1 (and not just with high probability), with query complexity $O(n \log n + \frac{k^2 \log n}{\mathcal{H}^2(f_+ \| f_-)})$. Since $f_+$ and $f_-$ can be arbitrary, not knowing the distributions provides a major challenge, and we believe, our recipe could be fruitful for designing further parameter-free algorithms. We note that all our algorithms are computationally efficient - in fact, the time required is bounded by the size of the side information matrix, i.e., $O(n^2)$.

**Theorem 1.** *Let the number of clusters $k$ be unknown and $f_+$ and $f_-$ be unknown discrete distributions with fixed cardinality of support. There exists an efficient (polynomial-time) Monte Carlo algorithm for Query-Cluster that has query complexity $O(\min(nk, \frac{k^2 \log n}{\mathcal{H}^2(f_+ \| f_-)}))$ and recovers all the clusters accurately with probability $1 - o(\frac{1}{n})$. Moreover there exists an efficient Las Vegas algorithm that with probability $1 - o(\frac{1}{n})$ has query complexity $O(n \log n + \min(nk, \frac{k^2 \log n}{\mathcal{H}^2(f_+ \| f_-)}))$.*

**Lower Bound.** Our main lower bound result is information theoretic, and can be summarized in the following theorem. Note especially that, for lower bound we can assume the knowledge of $k$, $f_+$, $f_-$ in contrast to upper bounds, which makes the results stronger. In addition, $f_+$ and $f_-$ can be discrete or continuous distributions. Note that when $\mathcal{H}^2(f_+ \| f_-)$ is close to 1, e.g., when the side information is perfect, no queries are required. However, that is not the case in practice, and we are interested in the region where $f_+$ and $f_-$ are "close", that is $\mathcal{H}^2(f_+ \| f_-)$ is small.

**Theorem 2.** *Assume $\mathcal{H}^2(f_+ \| f_-) \le \frac{1}{18}$. Any (possibly randomized) algorithm with the knowledge of $f_+$, $f_-$, and the number of clusters $k$, that does not perform $\Omega \left( \min \left\{ nk, \frac{k^2}{\mathcal{H}^2(f_+ \| f_-)} \right\} \right)$ expected*

*number of queries, will be unable to return the correct clustering with probability at least $\frac{1}{6} - O(\frac{1}{\sqrt{k}})$. And to recover the clusters with probability $1$, the number of queries must be $\Omega\Big(n + \min\{nk, \frac{k^2}{\mathcal{H}^2(f_+\|f_-)}\}\Big)$.*

The lower bound therefore matches the query complexity upper bound within a logarithmic factor.

Note that when no querying is allowed, this turns out exactly to be the setting of stochastic block model though with much general distributions. We have analyzed this case in Appendix C. To see how the probability of error must scale, we have used a generalized version of Fano's inequality (e.g., [23]). However, when the number of queries is greater than zero, plus when queries can be adaptive, any such standard technique fails. Hence, significant effort has to be put forth to construct a setting where information theoretic minimax bounds can be applied. This lower bound could be of independent interest, and provides a general framework for deriving lower bounds for fundamental problems of classification, hypothesis testing, distribution testing etc. in the interactive learning setting. They may also lead to new lower bound proving techniques in the related multi-round communication complexity model where information again gets revealed adaptively.

**Organization.** The proof of the lower bound is provided in Section 2. The Monte Carlo algorithm is given in Section 3. The detailed proof of the Monte Carlo algorithm, and the Las Vegas algorithm and its proof are given in Appendix A and Appendix B respectively in the supplementary material for space constraint.

## 2    Lower Bound (Proof of Theorem 2)

In this section, we develop our information theoretic lower bounds. We prove a more general result from which Theorem 2 follows.

**Lemma 1.** *Consider the case when we have $k$ equally sized clusters of size $a$ each (that is total number of elements is $n = ka$). Suppose we are allowed to make at most $Q$ adaptive queries to the oracle. The probability of error for any algorithm for* Query-Cluster *is at least,*

$$1 - \frac{2}{k}\Big(1 + \sqrt{\frac{4Q}{ak}}\Big)^2 - \frac{4Q}{ak(k-1)} - 2\sqrt{a}\mathcal{H}(f_+\|f_-).$$

The main high-level technique to prove Lemma 1 is the following. Suppose, a node is to be assigned to a cluster. This situation is obviously akin to a $k$-hypothesis testing problem, and we want to use a lower bound on the probability of error. The side information and the query answers constitute a random vector whose distributions (among the $k$ possible) must be far apart for us to successfully identify the clustering. But the main challenge comes from the interactive nature of the algorithm since it reveals deterministic information and into characterizing the set of elements that are not queried much by the algorithm.

*Proof of Lemma 1.* Since the total number of queries is $Q$, the average number of queries per element is at most $\frac{2Q}{ak}$. Therefore there exist at least $\frac{ak}{2}$ elements that are queried at most $T < \frac{4Q}{ak}$ times. Let $x$ be one such element. We just consider the problem of assignment of $x$ to a cluster (all other elements have been correctly assigned already), and show that any algorithm will make wrong assignment with positive probability.

**Step 1: Setting up the hypotheses.** Note that the side information matrix $W = (w_{i,j})$ is provided where the $w_{i,j}$s are independent random variables. Now assume the scenario when we use an algorithm ALG to assign $x$ to one of the $k$ clusters, $V_u, u = 1, \ldots, k$. Therefore, given $x$, ALG takes as input the random variables $w_{i,x}$s where $i \in \sqcup_t V_t$, makes some queries involving $x$ and outputs a cluster index, which is an assignment for $x$. Based on the observations $w_{i,x}$s, the task of ALG is thus a multi-hypothesis testing among $k$ hypotheses. Let $H_u, u = 1, \ldots k$ denote the $k$ different hypotheses $H_u : x \in V_u$. And let $P_u, u = 1, \ldots k$ denote the joint probability distributions of the random matrix $W$ when $x \in V_u$. In short, for any event $\mathcal{A}$, $P_u(\mathcal{A}) = \Pr(\mathcal{A}|H_u)$. Going forward, the subscript of probabilities or expectations will denote the appropriate conditional distribution.

**Step 2: Finding "weak" clusters.** There must exist $t \in \{1, \ldots, k\}$ such that,

$$\sum_{v=1}^{k} P_t\{ \text{ a query made by ALG involving cluster } V_v\} \leq \mathbb{E}_t\{\text{Number of queries made by ALG}\} \leq T.$$

We now find a subset of clusters, that are "weak," i.e., not queried enough if $H_t$ were true. Consider the set $J' \equiv \{v \in \{1, \ldots, k\} : P_t\{$ a query made by ALG involving cluster $V_v\} < \frac{2T}{k(1-\beta)}\}$, where $\beta \equiv \frac{1}{1+\sqrt{\frac{4Q}{ak}}}$. We must have, $(k - |J'|) \cdot \frac{2T}{k(1-\beta)} \leq T$, which implies, $|J'| \geq \frac{(1+\beta)k}{2}$.

Now, to output a cluster without using the side information, ALG has to either make a query to the actual cluster the element is from, or query at least $k - 1$ times. In any other case, ALG must use the side information (in addition to using queries) to output a cluster. Let $\mathcal{E}^u$ denote the event that ALG outputs cluster $V_u$ by using the side information. Let $J'' \equiv \{u \in \{1, \ldots, k\} : P_t(\mathcal{E}^u) \leq \frac{2}{\beta k}\}$. Since $\sum_{u=1}^{k} P_t(\mathcal{E}^u) \leq 1$, we must have, $(k - |J''|) \cdot \frac{2}{\beta k} < 1$, or $|J''| > k - \frac{\beta k}{2} = \frac{(2-\beta)k}{2}$. We have, $|J' \cap J''| > \frac{(1+\beta)k}{2} + \frac{(2-\beta)k}{2} - k = \frac{k}{2}$. This means, $\{V_u : u \in J' \cap J''\}$ contains more than $\frac{ak}{2}$ elements. Since there are $\frac{ak}{2}$ elements that are queried at most $T$ times, these two sets must have nonzero intersection. Hence, we can assume that $x \in V_\ell$ for some $\ell \in J' \cap J''$, i.e., let $H_\ell$ be the true hypothesis. Now we characterize the error events of the algorithm ALG in assignment of $x$.

**Step 3: Characterizing error events for "$x$".** We now consider the following two events. $\mathcal{E}_1 = \{$a query made by ALG involving cluster $V_\ell\}; \mathcal{E}_2 = \{k - 1$ or more queries were made by ALG$\}$.

Note that if the algorithm ALG can correctly assign $x$ to a cluster without using the side information then either of $\mathcal{E}_1$ or $\mathcal{E}_2$ must have to happen. Recall, $\mathcal{E}^\ell$ denotes the event that ALG outputs cluster $V_\ell$ using the side information. Now consider the event $\mathcal{E} \equiv \mathcal{E}^\ell \bigcup \mathcal{E}_1 \bigcup \mathcal{E}_2$. The probability of correct assignment is at most $P_\ell(\mathcal{E})$. We now bound this probability of correct recovery from above.

**Step 4: Bounding probability of correct recovery via Hellinger distance.** We have,
$$P_\ell(\mathcal{E}) \leq P_t(\mathcal{E}) + |P_\ell(\mathcal{E}) - P_t(\mathcal{E})| \leq P_t(\mathcal{E}) + \|P_\ell - P_t\|_{TV} \leq P_t(\mathcal{E}) + \sqrt{2}\mathcal{H}(P_\ell\|P_t),$$
where, $\|P - Q\|_{TV} \equiv \sup_A |P(A) - Q(A)|$ denotes the total variation distance between two probability distributions $P$ and $Q$ and in the last step we have used the relationship between total variation distance and the Hellinger divergence (see, for example, [38, Eq. (3)]). Now, recall that $P_\ell$ and $P_t$ are the joint distributions of the independent random variables $w_{i,x}, i \in \cup_u V_u$. Now, we use the fact that squared Hellinger divergence between product distribution of independent random variables are less than the sum of the squared Hellinger divergence between the individual distribution. We also note that the divergence between identical random variables are 0. We obtain
$$\sqrt{2\mathcal{H}^2(P_\ell\|P_t)} \leq \sqrt{2 \cdot 2a\mathcal{H}^2(f_+\|f_-)} = 2\sqrt{a}\mathcal{H}(f_+\|f_-).$$
This is true because the only times when $w_{i,x}$ differs under $P_t$ and under $P_\ell$ is when $x \in V_t$ or $x \in V_\ell$. As a result we have, $P_\ell(\mathcal{E}) \leq P_t(\mathcal{E}) + 2\sqrt{a}\mathcal{H}(f_+\|f_-)$. Now, using Markov inequality $P_t(\mathcal{E}_2) \leq \frac{T}{k-1} \leq \frac{4Q}{ak(k-1)}$. Therefore,
$$P_t(\mathcal{E}) \leq P_t(\mathcal{E}^\ell) + P_t(\mathcal{E}_1) + P_t(\mathcal{E}_2) \leq \frac{2}{\beta k} + \frac{8Q}{ak^2(1-\beta)} + \frac{4Q}{ak(k-1)}.$$

Therefore, putting the value of $\beta$ we get, $P_\ell(\mathcal{E}) \leq \frac{2}{k}\left(1 + \sqrt{\frac{4Q}{ak}}\right)^2 + \frac{4Q}{ak(k-1)} + 2\sqrt{a}\mathcal{H}(f_+\|f_-)$, which proves the lemma. $\qquad\square$

*Proof of Theorem 2.* Consider two cases. In the first case, suppose, $nk < \frac{k^2}{9\mathcal{H}^2(f_+\|f_-)}$. Now consider the situation of Lemma 1, with $a = \frac{n}{k}$. The probability of error of any algorithm must be at least,
$$1 - \frac{2}{k}\left(1 + \sqrt{\frac{4Q}{ak}}\right)^2 - \frac{4Q}{ak(k-1)} - \frac{2}{3} \geq \frac{1}{6} - O(\frac{1}{\sqrt{k}}),$$ if the number of queries $Q \leq \frac{nk}{72}$.

In the second case, suppose $nk \geq \frac{k^2}{9\mathcal{H}^2(f_+\|f_-)}$. Assume, $a = \lfloor \frac{1}{9\mathcal{H}^2(f_+\|f_-)} \rfloor$. Then $a \geq 2$, since $\mathcal{H}^2(f_+\|f_-) \leq \frac{1}{18}$. We have $nk \geq k^2 a$. Consider the situation when we are already given a complete cluster $V_k$ with $n - (k-1)a$ elements, remaining $(k-1)$ clusters each has 1 element, and the rest $(a-1)(k-1)$ elements are evenly distributed (but yet to be assigned) to the $k - 1$ clusters. Now we are exactly in the situation of Lemma 1 with $k - 1$ playing the role of $k$. If we have $Q < \frac{ak^2}{72}$, The probability of error is at least $1 - o_k(1) - \frac{1}{6} - \frac{2}{3} = \frac{1}{6} - O(\frac{1}{\sqrt{k}})$. Therefore $Q$ must be $\Omega(\frac{k^2}{\mathcal{H}^2(f_+\|f_-)})$. Note that in this proof we have not in particular tried to optimize the constants.

If we want to recover the clusters with probability 1, then $\Omega(n)$ is a trivial lower bound. Hence, coupled with the above we get a lower bound of $\Omega(n + \min\{nk, \frac{k^2}{\mathcal{H}^2(f_+\|f_-)}\})$ in that case. $\qquad\square$

# 3 Algorithms

We propose two algorithms (Monte Carlo and Las Vegas) both of which are completely parameter free that is they work without any knowledge of $k$, $f_+$ and $f_-$, and meet the respective lower bounds within an $O(\log n)$ factor. Here we present the Monte Carlo algorithm which drastically reduces the number of queries from $O(nk)$ (no side information) to $O(\frac{k^2 \log n}{\mathcal{H}^2(f_+ \| f_-)})$ and recovers the clusters exactly with probability at least $1 - o_n(1)$. The detailed proof of it, as well as the Las Vegas algorithm are presented in Appendix A and Appendix B respectively in the supplementary material.

Our algorithm uses a subroutine called Membership that takes as input an element $v \in V$ and a subset of elements $\mathcal{C} \subseteq V \setminus \{v\}$. Assume that $f_+$, $f_-$ are discrete distributions over fixed set of $q$ points $a_1, a_2, \ldots, a_q$; that is $w_{i,j}$ takes value in the set $\{a_1, a_2, \ldots, a_q\}$. Define the empirical "inter" distribution $p_{v,\mathcal{C}}$ for $i = 1, \ldots, q$, $p_{v,\mathcal{C}}(i) = \frac{|\{u \in \mathcal{C} : w_{u,v} = a_i\}|}{|\mathcal{C}|}$ Also compute the "intra" distribution $p_{\mathcal{C}}$ for $i = 1, \ldots, q$, $p_{\mathcal{C}}(i) = \frac{|\{(u,v) \in \mathcal{C} \times \mathcal{C} : u \neq v, w_{u,v} = a_i\}|}{|\mathcal{C}|(|\mathcal{C}|-1)}$. Then we use Membership$(v, \mathcal{C})$ $= -\mathcal{H}^2(p_{v,\mathcal{C}} \| p_{\mathcal{C}})$ as affinity of vertex $v$ to $\mathcal{C}$, where $\mathcal{H}(p_{v,\mathcal{C}} \| p_{\mathcal{C}})$ denotes the Hellinger divergence between distributions. Note that since the membership is always negative, a higher membership implies that the 'inter' and 'intra' distributions are closer in terms of Hellinger distance.

Designing a parameter free Monte Carlo algorithm seems to be highly challenging as here, the number of queries depends only logarithmically with $n$. Intuitively, if an element $v$ has the highest membership in some cluster $\mathcal{C}$, then $v$ should be queried with $\mathcal{C}$ first. Also an estimation from side information is reliable when the cluster already has enough members. Unfortunately, we know neither whether the current cluster size is reliable, nor we are allowed to make even one query per element.

To overcome this bottleneck, we propose an iterative-update algorithm which we believe will find more uses in developing parameter free algorithms. We start by querying a few points so that there is at least one cluster with $\Theta(\log n)$ points. Now based on these queried memberships, we learn two empirical distributions $p_+^1$ from intra-cluster similarity values, and $p_-^1$ from inter-cluster similarity values. Given an element $v$ which has not been clustered yet, and a cluster $\mathcal{C}$ with the highest number of current members, we would like to consider the submatrix of side information pertaining to $v$ and all $u \in \mathcal{C}$ and determine whether that side information is generated from $f_+$ or $f_-$. We know if the statistical distance between $f_+$ and $f_-$ is small, then we would need more members in $\mathcal{C}$ to successfully do this test. Since we do not know $f_+$ and $f_-$, we compute the squared Hellinger divergence between $p_+^1$ and $p_-^1$, and use that to compute a threshold $\tau_1$ on the size of $\mathcal{C}$. If $\mathcal{C}$ crosses this size threshold, we just use the side information to determine if $v$ should belong to $\mathcal{C}$. Otherwise, we query further until there is one cluster with size $\tau_1$, and re-estimate the empirical distributions $p_+^2$ and $p_-^2$. Again, we recompute a threshold $\tau_2$, and stop if the cluster under consideration crosses this new threshold. If not we continue. Interestingly, we can show when the process converges, we have a very good estimate of $\mathcal{H}(f_+ \| f_-)$ and, moreover it converges fast.

**Algorithm. Phase 1. Initialization.** We initialize the algorithm by selecting any element $v$ and creating a singleton cluster $\{v\}$. We then keep selecting new elements randomly and uniformly that have not yet been clustered, and query the oracle with it by choosing exactly one element from each of the clusters formed so far. If the oracle returns $+1$ to any of these queries then we include the element in the corresponding cluster, else we create a new singleton cluster with it. We continue this process until one cluster has grown to a size of $\lceil C \log n \rceil$, where $C$ is a constant.

**Phase 2. Iterative Update.** Let $\mathcal{C}_1, \mathcal{C}_2, \ldots \mathcal{C}_{l_x}$ be the set of clusters formed after the $x$th iteration for some $l_x \leq k$, where we consider Phase 1 as the 0-th iteration. We estimate

$$p_{+,x} = \frac{|\{u, v \in \mathcal{C}_i : u \neq v, w_{u,v} = a_i\}|}{\sum_{i=1}^{l_x} |\mathcal{C}_i|(|\mathcal{C}_i| - 1)}; p_{-,x} = \frac{|\{u \in \mathcal{C}_i, v \in \mathcal{C}_j, i < j, i, j \in [1, l_x] : w_{u,v} = a_i\}|}{\sum_{i=1}^{l_x} \sum_{i<j} |\mathcal{C}_i||\mathcal{C}_j|}$$

Define $M_x^E = \frac{C \log n}{\mathcal{H}(p_{+,x} \| p_{-,x})^2}$. If there is no cluster of size at least $M_x^E$ formed so far, we select a new element yet to be clustered and query it exactly once with the existing clusters (that is by selecting one arbitrary point from every cluster and querying the oracle with it and the new element), and include it in an existing cluster or create a new cluster with it based on the query answer. We then set $x = x + 1$ and move to the next iteration to get updated estimates of $p_{+,x}, p_{-,x}, M_x^E$ and $l_x$.

Else if there is a cluster of size at least $M_x^E$, we stop and move to the next phase.

**Phase 3. Processing the grown clusters.** Once Phase 2 has converged, let $p_+, p_-, \mathcal{H}(p_+\|p_-), M^E$ and $l$ be the final estimates. For every cluster $\mathcal{C}$ of size $|\mathcal{C}| \geq M^E$, call it grown and do the following.

(3A.) For every unclustered element $v$, if Membership$(v, \mathcal{C}) \geq -(\frac{4\mathcal{H}(p_+\|p_-)}{C} - \frac{2\mathcal{H}(p_+\|p_-)^2}{C\sqrt{\log n}})$, then we include $v$ in $\mathcal{C}$ *without* querying.

(3B.) We create a new list Waiting$(\mathcal{C})$, initially empty. If $-(\frac{4\mathcal{H}(p_+\|p_-)}{C} - \frac{2\mathcal{H}(p_+\|p_-)^2}{C\sqrt{\log n}}) >$ Membership$(v, \mathcal{C}) \geq -(\frac{4\mathcal{H}(p_+\|p_-)}{C} + \frac{2\mathcal{H}(p_+\|p_-)^2}{C\sqrt{\log n}})$, then we include $v$ in Waiting$(\mathcal{C})$. For every element in Waiting$(\mathcal{C})$, we query the oracle with it by choosing exactly one element from each of the clusters formed so far starting with $\mathcal{C}$. If oracle returns answer "yes" to any of these queries then we include the element in that cluster, else we create a new singleton cluster with it. We continue this until Waiting$(\mathcal{C})$ is exhausted.

We then call $\mathcal{C}$ completely grown, remove it from further consideration, and move to the next grown cluster. if there is no other grown cluster, then we move back to Phase 2.

**Analysis.** The main steps of the analysis are as follows (for full analysis see Appendix A).

1. First, Lemma 3 shows with high probability $\mathcal{H}(p_+\|p_-) \in [\mathcal{H}(f_+\|f_-) \pm \frac{4\mathcal{H}(p_+\|p_-)^2}{B\sqrt{\log n}}]$ for a suitable constant $B$ that depends on $C$. Using it, we can show the process converges whenever a cluster has grown to a size of $\frac{4C\log n}{\mathcal{H}^2(f_+\|f_-)}$. The proof relies on adapting the Sanov's Theorem (see Lemma 2) of information theory. We are measuring the distance between distributions via Hellinger distance, as opposed to KL divergence (which would have been a natural choice because of its presence in the rate function in Sanov's therem), because Hellinger distance is a metric which proves to be crucial in our analysis.
2. Lemma 5 and Corollary 1 show that every element that is included in $\mathcal{C}$ in Phase $(3A)$ truly belongs to $\mathcal{C}$, and elements that are not in Waiting$(\mathcal{C})$ can not be in $\mathcal{C}$ with high probability. Once Phase 2 has converged, if the condition of $(3A)$ is satisfied, the element must belong to $\mathcal{C}$. There is a small gray region of confidence interval $(3B)$ such that if an element belongs there, we cannot be sure either way, but if an element does not satisfy either $(3A)$ or $3B$, it cannot be part of $\mathcal{C}$.
3. Lemma 6 shows that size of Waiting$(\mathcal{C})$ is constant showing an anti-concentration property. This coupled with the fact that the process converges when a cluster reaches size $\frac{4C\log n}{\mathcal{H}^2(f_+\|f_-)}$ gives the desired query complexity bound in Lemma 7.

## 4 Experimental Results

In this section, we report experimental results on a popular bibliographic dataset `cora` [35] consisting of 1879 nodes, 191 clusters and 1699612 edges out of which 62891 are intra-cluster edges. We remove any singleton node from the dataset – the final number of vertices that we classify is 1812 with 124 clusters. We use the similarity function computation used by [18] to compute $f_+$ and $f_-$. The two distributions are shown in Figure 1 on the left. The Hellinger square divergence between the two distributions is 0.6. In order to observe the dependency of the algorithm performance on the learnt distributions, we perturb the exact distributions to obtain two approximate distributions as shown in Figure 1 (middle) with Hellinger square divergence being 0.4587. We consider three strategies. Suppose the cluster in which a node $v$ must be included has already been initialized and exists in the current solution. Moreover, suppose the algorithm decides to use queries to find membership of $v$. Then in the `best` strategy, only one query is needed to identify the cluster in which $v$ belongs. In the `worst` strategy, the algorithm finds the correct cluster after querying all the existing clusters whose current membership is not enough to take a decision using side information. In the `greedy` strategy, the algorithm queries the clusters in non-decreasing order of Hellinger square divergence between $f_+$ (or approximate version of it) and the estimated distribution from side information between $v$ and each existing clusters. Note that, in practice, we will follow the `greedy` strategy. Figure 2 shows the performance of each strategy. We plot the number of queries vs F1 Score which computes the harmonic mean of precision and recall. We observe that the performance of `greedy` strategy is very close to that of `best`. With just 1136 queries, `greedy` achieves 80% precision and close to 90% recall. The `best` strategy would need 962 queries to achieve that performance. The performance of our algorithm on the exact and approximate distributions are also very close which indicates it is enough to learn a distribution that is close to exact. For example, using the approximate distributions,

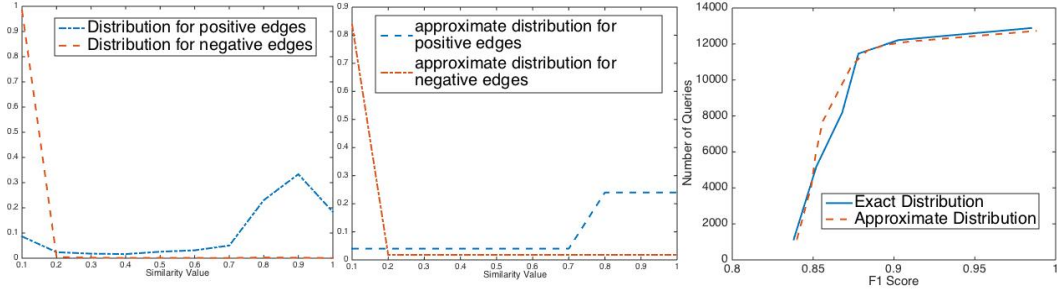

Figure 1: (left) Exact distributions of similarity values, (middle) approximate distributions of similarity values, (right) Number of Queries vs F1 Score for both distributions.

to achieve similar precision and recall, the `greedy` strategy just uses $1148$ queries, that is $12$ queries more than when we use when the distributions are known.

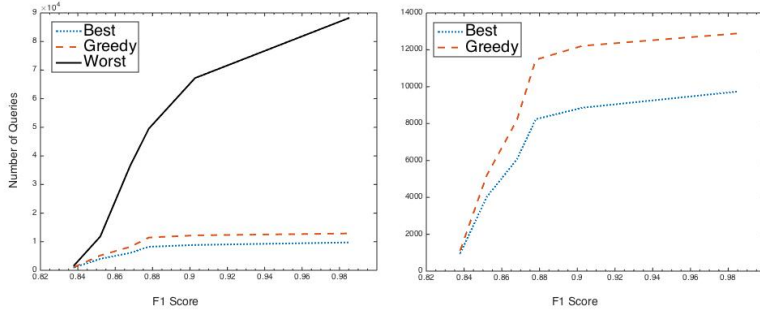

Figure 2: Number of Queries vs F1 Score using three strategies: best, greedy, worst.

**Discussion.** This is the first rigorous theoretical study of interactive clustering with side information, and it unveils many interesting directions for future study of both theoretical and practical significance (see Appendix D for more details). Having arbitrary $f_+$, $f_-$ is a generalization of SBM. Also it raises an important question about how SBM recovery threshold changes with queries. For sparse region of SBM, where $f_+$ is Bernoulli($\frac{a' \log n}{n}$) and $f_-$ is Bernoulli($\frac{b' \log n}{n}$), $a' > b'$, Lemma 1 is not tight yet. However, it shows the following trend. Let us set $a = \frac{n}{k}$ in Lemma 1 with the above $f_+, f_-$. We conjecture by ignoring the lower order terms and a $\sqrt{\log n}$ factor that with $Q$ queries, the sharp recovery threshold of sparse SBM changes from $(\sqrt{a'} - \sqrt{b'}) \geq \sqrt{k}$ to $(\sqrt{a'} - \sqrt{b'}) \geq \sqrt{k} \left(1 - \frac{Q}{nk}\right)$. Proving this bound remains an exciting open question.

We propose two computationally efficient algorithms that match the query complexity lower bound within $\log n$ factor and are completely parameter free. In particular, our iterative-update method to design Monte-Carlo algorithm provides a general recipe to develop any parameter-free algorithm, which are of extreme practical importance. The convergence result is established by extending Sanov's theorem from the large deviation theory which gives bound only in terms of KL-divergence. Due to the generality of the distributions, the only tool we could use is Sanov's theorem. However, Hellinger distance comes out to be the right measure both for lower and upper bounds. If $f_+$ and $f_-$ are common distributions like Gaussian, Bernoulli etc., then other concentration results stronger than Sanov may be applied to improve the constants and a logarithm factor to show the trade-off between queries and thresholds as in sparse SBM. While some of our results apply to general $f_{i,j}$s, a full picture with arbitrary $f_{i,j}$s and closing the gap of $\log n$ between the lower and upper bound remain an important future direction.

**Acknowledgement.** This work is supported in part by NSF awards CCF 1642658, CCF 1642550, CCF 1464310, CCF 1652303, a Yahoo ACE Award and a Google Faculty Research Award. We are particularly thankful to an anonymous reviewer whose comments led to notable improvement of the presentation of the paper.

## Footnotes

[1]Most recent works consider the region of interest as $p = \frac{a \log n}{n}$ and $q = \frac{b \log n}{n}$ for some $a > b > 0$.

[2]Our lower bound holds for continuous distributions as well.

[3]For simplicity of expression, we treat the sample space to be of constant size. However, all our results extend to any finite sample space scaling linearly with its size.

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
