[Supplementary Material]

# Query Complexity of Clustering with Side Information (Supplementary Material)

Arya Mazumdar and Barna Saha

## A  Upper Bound (Algorithm): Proof of Theorem 1 (Monte Carlo)

In this section, we first prove the correctness of the Monte Carlo algorithm, and in the subsequent section give the Las Vegas algorithm along with its proof.

One of the important tools that will be used in this section is Sanov's theorem from the large-deviation theory.

**Lemma 2** (Sanov's theorem). *Let $X_1, \ldots, X_n$ are iid random variables with a finite sample space $\mathcal{X}$ and distribution $P$. Let $P^n$ denote their joint distribution. Let $E$ be a set of probability distributions on $\mathcal{X}$. The empirical distribution $\tilde{P}_n$ gives probability $\tilde{P}_n(\mathcal{A}) = \frac{1}{n} \sum_{i=1}^{n} \mathbf{1}_{X_i \in \mathcal{A}}$ to any event $\mathcal{A}$. Then,*

$$P^n(\{x_1, \ldots, x_n\} : \tilde{P}_n \in E) \leq (n+1)^{|\mathcal{X}|} \exp(-n \min_{P^* \in E} D(P^* \| P)).$$

A continuous version of Sanov's theorem is also possible, especially when the set $E$ is convex (as a matter of fact the polynomial term in front of the right hand side can be omitted in cerain cases), but we omit here for clarity. The Sanov's theorem states, if we have an empirical distribution $P^n$ and a set of all distributions satisfying certain property $E$, then the probability $P^n \in E$ decreases exponentially with the minimum KL divergence of $P^n$ with any distribution in $E$. Note that, the KL divergence in the exponent of the Sanov's theorem naturally indicates an upper bound in terms of KL divergence. However, a major difficulty in dealing with KL divergence is that it is not a distance and does not satisfy triangle inequality. We overcome that by dealing with Hellinger distance instead.

The first step for the algorithm is to compute an approximation of $f_+$ and $f_-$, say $p_+$ and $p_-$ respectively by querying a few items. It is quite possible that $p_+$ and $p_-$ are crude approximations of $f_+$ and $f_-$ and working with them will lead to erroneous clustering. Interestingly, we show that by an iterative update and estimate process, we can obtain a $p_+$ and $p_-$ which are close to $f_+$ and $f_-$. Then we try to assign an element to a cluster by computing the empirical distribution from the side information matrix. There will be a grey region where we cannot be confident to either include or discard a vertex. We show that since the range of that region is small, the number of elements that will fall in the grey region is negligible, and we can query for them to resolve their true memberships.

### A.1  Monte Carlo Algorithm

We now design an algorithm that is completely parameter free, that is it has no knowledge of $k$, $f_+$, or $f_-$ and recovers all the clusters accurately with high probability[4]. The query complexity of the algorithm matches the worst case bound within an $O(\log n)$ factor. Note that, as a side information, we are given the noisy similarity matrix $W$.

Assume that $f_+, f_-$ are discrete distributions over $q$ points $a_1, a_2, \ldots, a_q$; that is $w_{i,j}$ takes value in the set $\{a_1, a_2, \ldots, a_q\}$. We will treat $q$ as a constant for simplicity of expression, otherwise all query complexity results scale by a factor of $q$. The algorithm uses a subroutine called Membership that takes as input an element $v \in V$ and a subset of elements $\mathcal{C} \subseteq V \setminus \{v\}$.

Compute the 'inter' distribution $p_{v,\mathcal{C}}$ for $i = 1, \ldots, q$, $p_{v,\mathcal{C}}(i) = \frac{1}{|\mathcal{C}|} \cdot |\{u \in \mathcal{C} : w_{u,v} = a_i\}|$.

Also compute the 'intra' distribution $p_{\mathcal{C}}$ for $i = 1, \ldots, q$, $p_{\mathcal{C}}(i) = \frac{1}{|\mathcal{C}|(|\mathcal{C}|-1)} \cdot |\{(u,v) \in \mathcal{C} \times \mathcal{C} : u \neq v, w_{u,v} = a_i\}|$. Then define Membership$(v, \mathcal{C}) = -\mathcal{H}(p_{v,\mathcal{C}} \| p_{\mathcal{C}})$. Note that, since the membership is always negative, a higher membership implies that the 'inter' and 'intra' distributions are closer in terms of the Hellinger distance.

The algorithm has several phases.

**Phase 1. Initialization.** We initialize the algorithm by selecting any vertex $v$ and creating a singleton cluster $\{v\}$. We then keep selecting new vertices randomly and uniformly that have not yet been clustered, and query the oracle with it by choosing exactly one vertex from each of the clusters formed so far. If the oracle returns $+1$ to any of these queries then we include the vertex in the corresponding cluster, else we create a new singleton cluster with it. We continue this process until at least one cluster has grown to a size of $\lceil C \log n \rceil$, where $C$ is an appropriately chosen constant[5] that depends on $q$.

**Observation 3.** *The number of queries made in Phase 1 is at most $O(k^2 \log n)$.*

*Proof.* We stop the process as soon as a cluster has grown to size of $\lceil C \log n \rceil$. Therefore, we may have clustered at most $k * \lceil C \log n \rceil$ vertices at this stage, each of which may have required $k$ queries to the oracle, one for every cluster. $\qquad\square$

**Phase 2. Iterative Update.** Let $\mathcal{C}_1, \mathcal{C}_2, ... \mathcal{C}_{l_x}$ be the set of clusters formed after the $x$th iteration for some $l_x \leq k$, where we consider Phase 1 as the 0-th iteration. We estimate

$$p_{+,x} = \frac{1}{\sum_{i=1}^{l_x} \binom{|\mathcal{C}_i|}{2}} \cdot |\{u, v \in \mathcal{C}_i : w_{u,v} = a_i\}|, \text{ and}$$

$$p_{-,x} = \frac{1}{\sum_{i=1}^{l_x} \sum_{j<i} |\mathcal{C}_i||\mathcal{C}_j|} \cdot |\{u \in \mathcal{C}_i, v \in \mathcal{C}_j, i < j, i, j \in [1, l_x] : w_{u,v} = a_i\}|$$

Define

$$M_x^E = \frac{C \log n}{\mathcal{H}(p_{+,x} \| p_{-,x})^2}.$$

If there is no cluster of size at least $M_x^E$ formed so far, we select a new vertex yet to be clustered and query it exactly once with the existing clusters (that is by selecting one arbitrary point from every cluster and querying the oracle with the new vertex and the selected one), and include it in an existing cluster or create a new cluster with it based on the query answer. We then set $x = x + 1$ and move to the next iteration to get updated estimates of $p_{+,x}, p_{-,x}, M_x^E$ and $l_x$.

Else if there is a cluster of size at least $M_x^E$, we stop and move to the next phase.

**Phase 3. Processing the grown clusters.** Once Phase 2 has converged, let $p_+, p_-, \mathcal{H}(p_+ \| p_-), M^E$ and $l$ be the final estimates. For every cluster $\mathcal{C}$ of size $|\mathcal{C}| \geq M^E$, we call it grown and we do the following.

(3A.) For every unclustered vertex $v$, if $\mathsf{Membership}(v, \mathcal{C}) \geq -(\frac{4\mathcal{H}(p_+\|p_-)}{C} - \frac{2\mathcal{H}(p_+\|p_-)^2}{C\sqrt{\log n}})$, then we include $v$ in $\mathcal{C}$ *without* querying.

(3B.) We create a new list $\mathsf{Waiting}(\mathcal{C})$, initially empty. If

$$-(\frac{4\mathcal{H}(p_+\|p_-)}{C} - \frac{2\mathcal{H}(p_+\|p_-)^2}{C\sqrt{\log n}}) > \mathsf{Membership}(v, \mathcal{C}) \geq -(\frac{4\mathcal{H}(p_+\|p_-)}{C} + \frac{2\mathcal{H}(p_+\|p_-)^2}{C\sqrt{\log n}}),$$

then we include $v$ in $\mathsf{Waiting}(\mathcal{C})$. For every vertex in $\mathsf{Waiting}(\mathcal{C})$, we query the oracle with it by choosing exactly one vertex from each of the clusters formed so far starting with $\mathcal{C}$. If oracle returns answer "yes" to any of these queries then we include the vertex in that cluster, else we create a new singleton cluster with it. We continue this until $\mathsf{Waiting}(\mathcal{C})$ is exhausted.

We then call $\mathcal{C}$ completely grown, remove it from further consideration, and move to the next grown cluster. if there is no other grown cluster, then we move back to Phase 2.

## A.2 Analysis

There are two parts to the analysis, showing the clusters are correct with high probability and determining the query complexity.

**Lemma 3.** *With probability at least* $1 - \frac{6}{n^3}$ *all of the following holds for an appropriately chosen constant* $B$

*(a)* $\mathcal{H}(p_+\|f_+) \leq \frac{2\mathcal{H}(p_+\|p_-)^2}{B\sqrt{\log n}}$

*(b)* $\mathcal{H}(p_-\|f_-) \leq \frac{2\mathcal{H}(p_+\|p_-)^2}{B\sqrt{\log n}}$

*(c)* $\mathcal{H}(p_+\|p_-)\left(1 + \frac{4\mathcal{H}(p_+\|p_-)}{B\sqrt{\log n}}\right) \geq \mathcal{H}(f_+\|f_-) \geq \mathcal{H}(p_+\|p_-)\left(1 - \frac{4\mathcal{H}(p_+\|p_-)}{B\sqrt{\log n}}\right)$

*Proof.* Let $\mathcal{C}$ be a cluster which according to the updated estimates of $p_+$ and $p_-$ has crossed the updated $M^E$ threshold. Since $|\mathcal{C}| \geq M^E$, $p_+$ is estimated based on at least $\binom{M^E}{2}$ edges. We assume the largest cluster size in the input instance is at most $\frac{n}{2}$[6]. Suppose the total number of vertices selected in Phase 1 and Phase 2 before $\mathcal{C}$ grew to $M^E$ is strictly less than $\frac{3M^E}{2}$. Then the expected number of vertices selected from $\mathcal{C}$ is at most $\frac{3M^E}{4}$. Then, by the Chernoff bound, the probability that the number of vertices selected from $\mathcal{C}$ is $M^E$ is at most $e^{-\frac{M^E}{36}}$. Taking $C \geq 118$, we get with probability at least $1 - \frac{1}{n^3}$, the number of vertices chosen from outside $\mathcal{C}$ is at least $\frac{M^E}{2}$. Thus, $p_-$ is estimated based on at least $\frac{(M^E)^2}{2}$ edges.

Here, we use the following version of the Chrenoff bound[7].

**Lemma 4** (The Chernoff Bound). *Let* $X_1, X_2, ...., X_n$ *be independent random variable taking values in* $\{0,1\}$ *with* $E[X_i] = p_i$. *Let* $X = \sum_{i=1}^n X_i$, *and* $\mu = E[X]$. *Then the following holds*

    *1. For* $0 < \delta \leq 1$, $Pr[X \leq (1-\delta)\mu] \leq e^{-\mu\delta^2/2}$

    *2. For* $0 < \delta \leq 1$, $Pr[X \geq (1+\delta)\mu] \leq e^{-\mu\delta^2/3}$

(a) Let $M = \binom{M^E}{2} \geq \frac{(M^E)^2}{3}$. Now, select $\delta = \sqrt{\frac{C'\log n}{M}}$, where $C'$ is a constant that ensures $n^{2C'} \geq n^{\frac{8\sqrt{C'}}{27\sqrt{3}}-6} \geq (M+1)^q \approx (M^E+1)^{2q}$, also $C' \geq 3$.

$$\Pr\left(\mathcal{H}(p_+\|f_+) \geq \delta\right) = f_+\left(\{p_+ : \mathcal{H}(p_+\|f_+) \geq \delta\right)$$
$$= (M+1)^q \exp(-M \min_{p:\mathcal{H}(p\|f_+)\geq\delta} D(p\|f_+)),$$

Here in the last step we have used Sanov's theorem (see, Lemma 2). Using the relationship between KL-divergence and Hellinger distance, we get

$$D(p\|f_+) \geq 2\mathcal{H}^2(p\|f_+) \geq 2\delta^2$$

where in the last step we used the optimization condition under the Sanov's theorem. Setting $\delta = \sqrt{\frac{C'\log n}{M}}$, $M \geq \frac{(M^E)^2}{3} = \frac{C^2\log^2 n}{3\mathcal{H}(p_+\|p_-)^4}$, we get $\delta = \frac{\sqrt{3C'}\mathcal{H}(p_+\|p_-)^2}{C\sqrt{\log n}}$. Let us take $B' = \frac{C}{\sqrt{3C'}}$, and $B = \sqrt{\frac{C}{C'}}$, we have $B \leq B'$ and we get

$$\Pr\left(\mathcal{H}(p_+\|f_+) \geq \frac{2\mathcal{H}(p_+\|p_-)^2}{B'\sqrt{\log n}}\right) \leq \frac{1}{n^3}$$

Hence,

$$\Pr\left(\mathcal{H}(p_+\|f_+) \geq \frac{2\mathcal{H}(p_+\|p_-)^2}{B\sqrt{\log n}}\right) \leq \frac{1}{n^3}$$

(b) Following a similar argument as above, we get

$$\Pr\left(\mathcal{H}(p_-\|f_-) \geq \frac{2\mathcal{H}(p_+\|p_-)^2}{B\sqrt{\log n}}\right) \leq \frac{1}{n^3}$$

(c) Now

$$\mathcal{H}(f_+\|f_-) \geq \mathcal{H}(p_+\|p_-) - \mathcal{H}(p_+\|f_+) - \mathcal{H}(p_-\|f_-) \quad \text{by applying triangle inequality}$$

$$\geq \mathcal{H}(p_+\|p_-) - \frac{4\mathcal{H}(p_+\|p_-)^2}{B\sqrt{\log n}} \quad \text{from (a) and (b) with probability at least } 1 - \frac{2}{n^3}$$

$$= \mathcal{H}(p_+\|p_-)\left(1 - \frac{4\mathcal{H}(p_+\|p_-)}{B\sqrt{\log n}}\right)$$

Similarly,

$$\mathcal{H}(p_+\|p_-) \geq \mathcal{H}(f_+\|f_-) - \mathcal{H}(p_+\|f_+) - \mathcal{H}(p_-\|f_-) \quad \text{by triangle inequality}$$

$$\geq \mathcal{H}(f_+\|f_-) - \frac{4\mathcal{H}(p_+\|p_-)^2}{B\sqrt{\log n}} \quad \text{from (a) and (b) with probability at least } 1 - \frac{2}{n^3}$$

Hence, by union bound all of (a), (b) and (c) hold with probability at least $1 - \frac{6}{n^3}$. $\qquad\square$

**Lemma 5.** *Let $\mathcal{C}$ be a cluster considered in Phase 3 of size at least $M^E$ then the following holds with probability at least $1 - o_n(1)$.*

*(a) If* $\mathsf{Membership}(v,\mathcal{C}) > -\left(\frac{\mathcal{H}(p_+\|p_-)}{B} - \frac{2\mathcal{H}(p_+\|p_-)^2}{B\sqrt{\log n}}\right)$ *then $v$ is in $\mathcal{C}$*

*(b) If $v \in \mathcal{C}$ then* $\mathsf{Membership}(v,\mathcal{C}) \geq -\left(\frac{\mathcal{H}(p_+\|p_-)}{B} + \frac{2\mathcal{H}(p_+\|p_-)^2}{B\sqrt{\log n}}\right)$

*Proof.* Suppose $v \in \mathcal{C}$. Then for any $\delta > 0$, we have

$$\Pr\left(\mathcal{H}(p_{v,\mathcal{C}}\|f_+) > \delta \mid v \in \mathcal{C}\right) = f_+\left(\mathcal{H}(p_{v,\mathcal{C}}\|f_+) > \delta\right)$$

$$\leq (M^E+1)^q \exp(-M^E \min_{p:\mathcal{H}(p\|f_+)\geq\delta} D(p\|f_+)) \quad \text{(by Sanov's theorem)}$$

$$\leq (M^E+1)^q \exp(-M^E \min_{p:\mathcal{H}(p\|f_+)\geq\delta} 2\mathcal{H}^2(p\|f_+))$$

(noting the relationship between KL-divergence and Hellinger distance)

$$\leq (M^E+1)^q \exp(-2M^E\delta^2)$$

Setting $M^E\delta^2 = C'\log n$, we get $\delta = \sqrt{\frac{C'\log n}{M^E}} = \sqrt{\frac{C'}{C}}\mathcal{H}(p_+\|p_-) = \frac{\mathcal{H}(p_+\|p_-)}{B}$ (by noting the value of $B$), we get

$$\Pr\left(\mathcal{H}(p_{v,\mathcal{C}}\|f_+) > \frac{\mathcal{H}(p_+\|p_-)}{B} \mid v \in \mathcal{C}\right) \leq \frac{1}{n^3} \quad \text{(by noting the value of } C')$$

Similarly,

$$\Pr\left(\mathcal{H}(p_{v,\mathcal{C}}\|f_-) > \frac{\mathcal{H}(p_+\|p_-)}{B} \mid v \notin \mathcal{C}\right) \leq \frac{1}{n^3}$$

Therefore, with at least $1 - \frac{2}{n^2}$ probability (by applying union bound over all $v$ the following hold.
(i) If $v \in \mathcal{C}$ then $\mathcal{H}(p_{v,\mathcal{C}}\|f_+) < \frac{\mathcal{H}(p_+\|p_-)}{B}$ and (ii) If $v \notin \mathcal{C}$ then $\mathcal{H}(p_{v,\mathcal{C}}\|f_-) < \frac{\mathcal{H}(p_+\|p_-)}{B}$.

(a) We have $\mathsf{Membership}(v,\mathcal{C}) > -\left(\frac{\mathcal{H}(p_+\|p_-)}{B} - \frac{2\mathcal{H}(p_+\|p_-)^2}{B\sqrt{\log n}}\right)$, that is $\mathcal{H}(p_{v,\mathcal{C}}\|p_+) < \frac{\mathcal{H}(p_+\|p_-)}{B} - \frac{2\mathcal{H}(p_+\|p_-)^2}{B\sqrt{\log n}}$. Suppose if possible $v \notin \mathcal{C}$. Then, we have

$$\mathcal{H}(p_{v,\mathcal{C}}\|f_+) \leq \mathcal{H}(p_{v,\mathcal{C}}\|p_+) + \mathcal{H}(p_+\|f_+) \quad \text{by triangle inequality}$$

$$< \frac{\mathcal{H}(p_+\|p_-)}{B} - \frac{2\mathcal{H}(p_+\|p_-)^2}{B\sqrt{\log n}} + \mathcal{H}(p_+\|f_+) \quad \text{applying condition on } \mathsf{Membership}(v, \mathcal{C})$$

$$\leq \frac{\mathcal{H}(p_+\|p_-)}{B} \quad \text{from Lemma 3 (a) with probability at least } 1 - \frac{1}{n^3}$$

Then we have,

$$\mathcal{H}(p_{v,\mathcal{C}}\|f_-) \geq \mathcal{H}(f_+\|f_-) - \mathcal{H}(p_{v,\mathcal{C}}\|f_+) \quad \text{by triangle inequality}$$

$$\geq \mathcal{H}(p_+\|p_-) - \frac{4\mathcal{H}(p_+\|p_-)^2}{B\sqrt{\log n}} - \mathcal{H}(p_{v,\mathcal{C}}\|f_+) \quad \text{from Lemma 3 (c) with probability at least } 1 - \frac{2}{n^3}$$

$$\geq \left(1 - \frac{1}{B}\right)\mathcal{H}(p_+\|p_-) - \frac{4\mathcal{H}(p_+\|p_-)^2}{B\sqrt{\log n}} \quad \text{with probability at least } 1 - \frac{3}{n^3}$$

$$\geq \left(1 - \frac{1}{B} - \frac{4}{B\sqrt{\log n}}\right)\mathcal{H}(p_+\|p_-) \quad \text{since } \mathcal{H}(p_+\|p_-) \leq 1$$

$$> \frac{\mathcal{H}(p_+\|p_-)}{B} \quad \text{by taking } B > 6, \text{ or } C \geq 36C'$$

This contradicts that $v \notin \mathcal{C}$.

(b) Now assume $v \in \mathcal{C}$ but $\mathsf{Membership}(v, \mathcal{C}) \geq -(\frac{\mathcal{H}(p_+\|p_-)}{B} + \frac{2\mathcal{H}(p_+\|p_-)^2}{B\sqrt{\log n}})$, that is $\mathcal{H}(p_{v,\mathcal{C}}\|p_+) \geq \frac{\mathcal{H}(p_+\|p_-)}{B} + \frac{2\mathcal{H}(p_+\|p_-)^2}{B\sqrt{\log n}}$. We have

$$\mathcal{H}(p_{v\mathcal{C}}\|f_+) \geq \mathcal{H}(p_{v\mathcal{C}}\|p_+) - \mathcal{H}(f_+\|p_+)$$

$$\geq \frac{\mathcal{H}(p_+\|p_-)}{B} + \frac{2\mathcal{H}(p_+\|p_-)^2}{B\sqrt{\log n}} - \mathcal{H}(f_+\|p_+) \quad \text{applying condition on } \mathsf{Membership}(v, \mathcal{C})$$

$$\geq \frac{\mathcal{H}(p_+\|p_-)}{B} \quad \text{from Lemma 3 (a) with probability at least } 1 - \frac{1}{n^3}$$

This contradicts the fact that $v \in \mathcal{C}$. $\qquad\square$

**Corollary 1.** *Let $\mathcal{C}$ be a cluster considered in Phase 3 of size at least $M^E$ then the following hold with probability at least $1 - \frac{2}{n^2}$.*

*(a) Vertices that are included in $\mathcal{C}$ in Phase $(3A)$ truly belong to $\mathcal{C}$.*

*(b) Vertices that are not in $\mathsf{Waiting}(\mathcal{C})$ can not be in $\mathcal{C}$.*

*Proof.* Follows from Lemma 5 (a) and (b) respectively. $\qquad\square$

**Lemma 6.** *Let $\mathcal{C}$ be a cluster considered in Phase 3 of size at least $M^E$ and $\hat{\mathcal{C}}$ denotes the true cluster with $\mathcal{C} \subseteq \hat{\mathcal{C}}$. Then after Phase $(3A)$, $|\hat{\mathcal{C}} \setminus \mathcal{C}| = o(1)$ with probability at least $1 - \frac{1}{n^2}$.*

*Proof.* We have from Lemma 5 that for $v$ to belong to $\hat{\mathcal{C}}$, it must satisfy $\mathsf{Membership}(v, \mathcal{C}) \geq -(\frac{\mathcal{H}(p_+\|p_-)}{B} + \frac{2\mathcal{H}(p_+\|p_-)^2}{B\sqrt{\log n}})$. On the otherhand, if $v$ has $\mathsf{Membership}(v, \mathcal{C}) > -(\frac{\mathcal{H}(p_+\|p_-)}{B} - \frac{2\mathcal{H}(p_+\|p_-)^2}{B\sqrt{\log n}})$ then $v$ has already been included in $\mathcal{C}$. Therefore, the grey region of $\mathsf{Membership}(v, \mathcal{C})$ values for which we cannot decide on whether or not to include $v$ to $\mathcal{C}$ is when $\mathsf{Membership}(v, \mathcal{C}) \in -\frac{\mathcal{H}(p_+\|p_-)}{B} \pm \frac{2\mathcal{H}(p_+\|p_-)^2}{B\sqrt{\log n}}$, that is $\mathcal{H}(p_{v,\mathcal{C}}\|p_+) \in \frac{\mathcal{H}(p_+\|p_-)}{B} \pm \frac{2\mathcal{H}(p_+\|p_-)^2}{B\sqrt{\log n}}$.

Now,

$$\Pr\left(\mathcal{H}(p_{v,\mathcal{C}}\|p_+) \in \frac{\mathcal{H}(p_+\|p_-)}{B} \pm \frac{2\mathcal{H}(p_+\|p_-)^2}{B\sqrt{\log n}}\right) \leq \Pr\left(\mathcal{H}(p_{v,\mathcal{C}}\|p_+) \geq \frac{\mathcal{H}(p_+\|p_-)}{B} - \frac{2\mathcal{H}(p_+\|p_-)^2}{B\sqrt{\log n}}\right)$$

$$\leq (M^E + 1)^q \exp\left(-M^E \min_{p:\mathcal{H}(p\|p_+)\geq \frac{\mathcal{H}(p_+\|p_-)}{B} - \frac{2\mathcal{H}(p_+\|p_-)^2}{B\sqrt{\log n}}} D(p\|f_+)\right) \quad \text{by Sanov's theorem}$$

Now,

$$D(p\|f_+) \geq 2\mathcal{H}(p\|f_+)^2 \geq 2\Big(\mathcal{H}(p\|p_+) - \mathcal{H}(p_+\|f_+)\Big)^2 \quad \text{by triangle inequality}$$

$$\geq 2\Big(\frac{\mathcal{H}(p_+\|p_-)}{B} - \frac{2\mathcal{H}(p_+\|p_-)^2}{B\sqrt{\log n}} - \mathcal{H}(p_+\|f_+)\Big)^2 \quad \text{from the optimization condition}$$

$$\geq 2\Big(\frac{\mathcal{H}(p_+\|p_-)}{B} - \frac{4\mathcal{H}(p_+\|p_-)^2}{B\sqrt{\log n}}\Big)^2 \quad \text{from Lemma 3 (a) with probability at least } 1 - \frac{1}{n^3}$$

$$= \frac{2\mathcal{H}^2(p_+\|p_-)}{B^2}\Big(1 - \frac{4\mathcal{H}(p_+\|p_-)}{B}\Big)^2 \geq \frac{2\mathcal{H}^2(p_+\|p_-)}{B^2}\Big(1 - \frac{4}{B}\Big)^2$$

$$\geq \frac{2\mathcal{H}^2(p_+\|p_-)}{27B} \quad \text{by inserting the minimum value for } \frac{1}{B}\Big(1 - \frac{4}{B}\Big)^2$$

Now $M^E \geq \frac{C\log n}{\mathcal{H}(p_+\|p_-)^2}$. Hence,

$$\Pr\Big(\mathcal{H}(p_{v,\mathcal{C}}\|p_+) \in \frac{\mathcal{H}(p_+\|p_-)}{B} \pm \frac{2\mathcal{H}(p_+\|p_-)^2}{B\sqrt{\log n}}\Big)$$

$$\leq (M^E + 1)^q \exp(-\frac{2C}{27B}\log n) + \frac{1}{n^3} = (M^E + 1)^q \exp(-\frac{4\sqrt{C'}}{27\sqrt{3}}\log n) + \frac{1}{n^3} \leq \frac{2}{n^3}$$

Hence the expected number of vertices $v \in \mathcal{C}$ in the grey region is $\leq \frac{2}{n^2}$. Thus by simple Markov inequality, after Phase $(3A)$, the probability that $|\hat{\mathcal{C}} \setminus \mathcal{C}| \geq 4$ is at most $\frac{1}{2n^2}$. Hence, with probability at least $1 - \frac{1}{2n^2}$, the size is bounded by 4.

$\square$

**Lemma 7.** *The algorithm asks at most $O(\frac{k^2 \log n}{\mathcal{H}(f_+\|f_-)^2})$ queries over the three phases with probability $1 - o_n(1)$.*

*Proof.* In Phase 1, as seen from Observation 3, the number of queries is $O(k^2 \log n) \leq O(\frac{k^2 \log n}{\mathcal{H}(f_+\|f_-)^2})$, as $0 \leq \mathcal{H}(f_+\|f_-)^2 \leq 1$.

In Phase 2, from Lemma 3, at any time when we have a grown cluster

$$\mathcal{H}(p_+\|p_-) \geq \mathcal{H}(f_+\|f_-) - \mathcal{H}(p_+\|f_+) - \mathcal{H}(p_-\|f_-) \quad \text{by triangle inequality}$$

$$\geq \mathcal{H}(f_+\|f_-) - \frac{4\mathcal{H}(p_+\|p_-)^2}{B\sqrt{\log n}} \quad \text{from Lemma 3}$$

Therefore,

$$\mathcal{H}(p_+\|p_-) \geq \frac{\mathcal{H}(f_+\|f_-)}{1 + \frac{4\mathcal{H}(p_+\|p_-)}{B\sqrt{\log n}}} \geq \frac{\mathcal{H}(f_+\|f_-)}{2}$$

This also shows whenever one cluster has grown to a size of $\frac{4C\log n}{\mathcal{H}^2(f_+\|f_-)}$, then $M^E$ must cross the threshold based on the newest estimate of $p_+$ and $p_-$. Hence, Phase 2 never grows a cluster beyond a size of $O(\frac{\log n}{\mathcal{H}^2(f_+\|f_-)})$ with probability $1 - \frac{1}{n^3}$. Hence, in Phase 2, the total number of queries can be at most $O\Big(\frac{k^2 \log n}{\mathcal{H}^2(f_+\|f_-)}\Big)$.

In Phase 3, the total number of queries made is at most $O(k^2)$ with probability at least $1 - \frac{1}{2n}$ due to Lemma 6, and applying union bound over all the clusters.

Thus, we get the overall query complexity is $O(\frac{k^2 \log n}{\mathcal{H}(f_+\|f_-)^2})$ with probability $1 - o_n(1)$, where $o_n(1)$ denotes a function of $n$ that goes to 0 with $n$. $\square$

Putting together all the lemmas, we arrive at the statement of Theorem 1.

# B   A Las Vegas Algorithm for Query-Cluster with an Oracle

While our lower bound results assume knowledge of $k$, $f_+$ and $f_-$, our algorithms, both Las Vegas and Monte Carlo versions, do not require any knowledge of these. In this section, we design a Las Vegas algorithm for clustering with oracle.

We do not know $k$, $f_+$, $f_-$, $\mu_+$, or $\mu_-$, and our goal is to design an algorithm with optimum query complexity for exact reconstruction of the clusters with probability 1. We are provided with the side information matrix $W = (w_{i,j})$ as an input.

Recall that, our algorithm uses a subroutine called Membership that takes as input an element $v \in V$ and a subset of elements (cluster) $\mathcal{C} \subseteq V \setminus \{v\}$. Assume that $f_+$, $f_-$ are discrete distributions over $q$ points $a_1, a_2, \ldots, a_q$; that is $w_{i,j}$ takes value in the set $\{a_1, a_2, \ldots, a_q\}$. We defined the empirical "inter" distribution $p_{v,\mathcal{C}}$ for $i = 1, \ldots, q$, $p_{v,\mathcal{C}}(i) = \frac{1}{|\mathcal{C}|} \cdot |\{u \in \mathcal{C} : w_{u,v} = a_i\}|$. Also compute the "intra" distribution $p_{\mathcal{C}}$ for $i = 1, \ldots, q$, $p_{\mathcal{C}}(i) = \frac{1}{|\mathcal{C}|(|\mathcal{C}|-1)} \cdot |\{(u, v) \in \mathcal{C} \times \mathcal{C} : u \neq v, w_{u,v} = a_i\}|$. Then we use Membership$(v, \mathcal{C}) = -\mathcal{H}^2(p_{v,\mathcal{C}} \| p_{\mathcal{C}})$ as affinity of vertex $v$ to cluster $\mathcal{C}$, where $\mathcal{H}(p_{v,\mathcal{C}} \| p_{\mathcal{C}})$ denotes the Hellinger divergence between distributions. Note that since the membership is always negative, a higher membership implies that the 'inter' and 'intra' distributions are closer in terms of Hellinger distance.

The algorithm works as follows. Let $\mathcal{C}_1, \mathcal{C}_2, ..., \mathcal{C}_l$ be the current clusters in nonincreasing order of size. We find the minimum index $j \in [1, l]$ such that there exists a vertex $v$ not yet clustered, with the highest average membership to $\mathcal{C}_j$, that is Membership$(v, \mathcal{C}_j) \geq$ Membership$(v, \mathcal{C}_{j'})$, $\forall j' \neq j$, and $j$ is the smallest index for which such a $v$ exists. We first check if $v \in \mathcal{C}_j$ by querying $v$ with any current member of $\mathcal{C}_j$. If not, then we group the clusters $\mathcal{C}_1, \mathcal{C}_2, .., \mathcal{C}_{j-1}$ in at most $\lceil \log n \rceil$ groups such that clusters in group $i$ has size in the range $[\frac{|\mathcal{C}_1|}{2^{i-1}}, \frac{|\mathcal{C}_1|}{2^i})$. For each group, we pick the cluster which has the highest average membership with respect to $v$, and check by querying whether $v$ belongs to that cluster. Even after this, if the membership of $v$ is not resolved, then we query $v$ with one member of each of the clusters that we have not checked with previously. If $v$ is still not clustered, then we create a new singleton cluster with $v$ as its sole member.

The pseudocode of the algorithm is given in Figure 3 We now give a proof of the Las Vegas part of Theorem 1 here using Algorithm 3. We crucially use the following lemma which proves a strong concentration inequality adapting the Sanov's Theorem (see Lemma 2) of information theory.

**Lemma 8.** *Suppose, $\mathcal{C}, \mathcal{C}' \subseteq V$, $\mathcal{C} \cap \mathcal{C}' = \emptyset$ and $|\mathcal{C}| \geq M$, $|\mathcal{C}'| \geq M = \frac{32 \log n}{\mathcal{H}^2(f_+ \| f_-)}$. Then,*

$$\Pr\left(\text{Membership}(v, \mathcal{C}') \geq \text{Membership}(v, \mathcal{C}) \mid v \in \mathcal{C}\right) \leq \frac{2}{n^3}.$$

*Proof.* Let $\beta = \frac{\mathcal{H}(f_+ \| f_-)}{2}$. If Membership$(v, \mathcal{C}') \geq$ Membership$(v, \mathcal{C})$ then we must have, $\mathcal{H}(p_{v,\mathcal{C}'} \| p_{\mathcal{C}'}) \leq \mathcal{H}(p_{v,\mathcal{C}} \| p_{\mathcal{C}})$. This means either $\mathcal{H}(p_{v,\mathcal{C}'} \| p_{\mathcal{C}'}) \leq \frac{\beta}{2}$ or $\mathcal{H}(p_{v,\mathcal{C}} \| p_{\mathcal{C}}) \geq \frac{\beta}{2}$. Now, using triangle inequality,

$$\Pr\left(\mathcal{H}(p_{v,\mathcal{C}'} \| p_{\mathcal{C}'}) \leq \frac{\beta}{2}\right) \leq \Pr\left(\mathcal{H}(p_{v,\mathcal{C}'} \| f_+) - \mathcal{H}(p_{\mathcal{C}'} \| f_+) \leq \frac{\beta}{2}\right)$$

$$\leq \Pr\left(\mathcal{H}(p_{v,\mathcal{C}'} \| f_+) \leq \beta \text{ or } \mathcal{H}(p_{\mathcal{C}'} \| f_+) \geq \frac{\beta}{2}\right) \leq \Pr\left(\mathcal{H}(p_{v,\mathcal{C}'} \| f_+) \leq \beta\right) + \Pr\left(\mathcal{H}(p_{\mathcal{C}'} \| f_+) \geq \frac{\beta}{2}\right).$$

Similarly,

$$\Pr\left(\mathcal{H}(p_{v,\mathcal{C}} \| p_{\mathcal{C}}) \geq \frac{\beta}{2}\right) \leq \Pr\left(\mathcal{H}(p_{v,\mathcal{C}} \| f_+) + \mathcal{H}(p_{\mathcal{C}} \| f_+) \geq \frac{\beta}{2}\right)$$

$$\leq \Pr\left(\mathcal{H}(p_{v,\mathcal{C}} \| f_+) \geq \frac{\beta}{4} \text{ or } \mathcal{H}(p_{\mathcal{C}} \| f_+) \geq \frac{\beta}{4}\right) \leq \Pr\left(\mathcal{H}(p_{v,\mathcal{C}} \| f_+) \geq \frac{\beta}{4}\right) + \Pr\left(\mathcal{H}(p_{\mathcal{C}} \| f_+) \geq \frac{\beta}{4}\right).$$

Now, using Sanov's theorem (Lemma 2), we have,

$$\Pr\left(\mathcal{H}(p_{v,\mathcal{C}'} \| f_+) \leq \beta\right) \leq (M + 1)^q \exp(-M \min_{p: \mathcal{H}(p \| f_+) \leq \beta} D(p \| f_-)).$$

At the optimizing $p$ of the exponent,

$$D(p \| f_-) \geq 2\mathcal{H}^2(p \| f_-) \qquad\qquad \text{relation between Hellinger and KL [38]}$$

Figure 3: Pseudocode: Las Vegas Algorithm for Query-Cluster

---

**Algorithm 1** Query-Cluster with Side Information. Input: $\{V, W\}$ (Note: $\mathcal{O}$ is the perfect oracle.

---

    ▷ *Initialization.*
1: Pick an arbitrary element $v$ and create a new cluster $\{v\}$. Set $V = V \setminus v$
2: **while** $V \neq \emptyset$ **do**
    ▷ *Let the number of current clusters be $l \geq 1$*
3:     Order the existing clusters in nonincreasing size.
    ▷ *Let $|\mathcal{C}_1| \geq |\mathcal{C}_2| \geq \ldots \geq |\mathcal{C}_l|$ be the ordering (w.l.o.g).*
4:     **for** $j = 1$ to $l$ **do**
5:         If $\exists v \in V$ such that $j = \max_{i \in [1,l]} \mathsf{Membership}(v, \mathcal{C}_i)$, then select $v$ and Break;
6:     **end for**
7:     $\mathcal{O}(v, u)$ where $u \in \mathcal{C}_j$
8:     **if** $\mathcal{O}(v, u) == $ " $+ 1$" **then**
9:         Include $v$ in $\mathcal{C}_j$. $V = V \setminus v$
10:     **else**
    ▷ *logarithmic search for membership in the large groups. Note $s \leq \lceil \log n \rceil$*
11:         Group $\mathcal{C}_1, \mathcal{C}_2, ..., \mathcal{C}_{j-1}$ into $s$ consecutive classes $H_1, H_2, ..., H_s$ such that the clusters in group $H_i$ have their current sizes in the range $[\frac{|\mathcal{C}_1|}{2^{i-1}}, \frac{|\mathcal{C}_1|}{2^i})$
12:         **for** $i = 1$ to $s$ **do**
13:             $j = \max_{a:\mathcal{C}_a \in H_i} \mathsf{Membership}(v, \mathcal{C}_a)$
14:             $\mathcal{O}(v, u)$ where $u \in \mathcal{C}_j$.
15:             **if** $\mathcal{O}(v, u) == $ " $+ 1$" **then**
16:                 Include $v$ in $\mathcal{C}_j$. $V = V \setminus v$. Break.
17:             **end if**
18:         **end for**
    ▷ *exhaustive search for membership in the remaining groups*
19:         **if** $v \in V$ **then**
20:             **for** $i = 1$ to $l + 1$ **do**
21:                 **if** $i = l + 1$ **then**         ▷ *v does not belong to any of the existing clusters*
22:                     Create a new cluster $\{v\}$. Set $V = V \setminus v$
23:                 **else**
24:                     **if** $\nexists u \in \mathcal{C}_i$ such that $(u, v)$ has already been queried **then**
25:                         $\mathcal{O}(v, u)$
26:                         **if** $\mathcal{O}(v, u) == $ " $+ 1$" **then**
27:                             Include $v$ in $\mathcal{C}_j$. $V = V \setminus v$. Break.
28:                         **end if**
29:                   **end if**
30:                 **end if**
31:             **end for**
32:         **end if**
33:     **end if**
34: **end while**

---

$$\geq 2(\mathcal{H}(f_+\|f_-) - \mathcal{H}(p\|f_+))^2 \qquad \text{from using triangle inequality}$$
$$\geq 2(2\beta - \beta)^2 \qquad \text{from noting the value of } \beta$$
$$= \frac{\mathcal{H}^2(f_+\|f_-)}{2}.$$

Again, using Sanov's theorem (Lemma 2), we have,

$$\Pr\left(\mathcal{H}(p_{\mathcal{C}'}\|f_+) \geq \frac{\beta}{2}\right) \leq (M+1)^q \exp(-M \min_{p:\mathcal{H}(p\|f_+) \geq \frac{\beta}{2}} D(p\|f_+)).$$

At the optimizing $p$ of the exponent,

$$D(p\|f_+) \geq 2\mathcal{H}^2(p\|f_+) \qquad \text{relation between Hellinger and KL divergences [38]}$$
$$\geq \frac{\beta^2}{2} \qquad \text{from noting the value of } \beta$$
$$= \frac{\mathcal{H}^2(f_+\|f_-)}{8}.$$

Now substituting this in the exponent, using the value of $M$ and doing the same exercise for the other two probabilities we get the claim of the lemma. $\qquad\square$

*Proof of Theorem 1, Las Vegas Algorithm.* First, The algorithm never includes a vertex in a cluster without querying it with at least one member of that cluster. Therefore, the clusters constructed by our algorithm are always proper subsets of the original clusters. Moreover, the algorithm never creates a new cluster with a vertex $v$ before first querying it with all the existing clusters. Hence, it is not possible that two clusters produced by our algorithm can be merged.

Let $\mathcal{C}_1, \mathcal{C}_2, ..., \mathcal{C}_l$ be the current non-empty clusters that are formed by Algorithm 3, for some $l \leq k$. Note that Algorithm 3 does not know $k$. Let without loss of generality $|\mathcal{C}_1| \geq |\mathcal{C}_2| \geq ... \geq |\mathcal{C}_l|$. Let there exists an index $i \leq l$ such that $|\mathcal{C}_1| \geq |\mathcal{C}_2| \geq \cdots \geq |\mathcal{C}_i| \geq M$, where $M = \frac{32 \log n}{\mathcal{H}^2(f_+\|f_-)}$. Of course, the algorithm does not know either $i$ or $M$. If even $|\mathcal{C}_1| < M$, then $i = 0$. Suppose $j'$ is the minimum index such that there exists a vertex $v$ with highest average membership in $\mathcal{C}_{j'}$. There are few cases to consider based on $j' \leq i$, or $j' > i$ and the cluster that truly contains $v$.

*Case 1.* $v$ *truly belongs to* $\mathcal{C}_{j'}$. In that case, we just make one query between $v$ and an existing member of $\mathcal{C}_{j'}$ and the first query is successful.

*Case 2.* $j' \leq i$ *and* $v$ *belongs to* $\mathcal{C}_j, j \neq j'$ *for some* $j \in \{1, ..., i\}$. Here we have Membership$(v, \mathcal{C}_{j'}) \geq$ Membership$(v, \mathcal{C}_j)$. Since both $\mathcal{C}_j$ and $\mathcal{C}_{j'}$ have at least $M$ current members, then using Lemma 8, this happens with probability at most $\frac{2}{n^3}$. Therefore, the number of queries involving $v$ before its membership gets determined is $\leq 1$ with probability at least $1 - \frac{2k}{n^3}$.

*Case 4.* $v$ *belongs to* $\mathcal{C}_j, j \neq j'$ *for some* $j > i$. In this case the algorithm may make $k$ queries involving $v$ before its membership gets determined.

*Case 5.* $j' > i$, *and* $v$ *belongs to* $\mathcal{C}_j$ *for some* $j \leq i$. In this case, there exists no $v$ with its highest membership in $\mathcal{C}_1, \mathcal{C}_2, ..., \mathcal{C}_i$.

Suppose $\mathcal{C}_1, \mathcal{C}_2, ..., \mathcal{C}_{j'}$ are contained in groups $H_1, H_2, ..., H_s$ where $s \leq \lceil \log n \rceil$. Let $\mathcal{C}_j \in H_t$, $t \in [1, s]$. Therefore, $|\mathcal{C}_j| \in [\frac{|\mathcal{C}_1|}{2^{t-1}}, \frac{|\mathcal{C}_1|}{2^t}]$. If $|\mathcal{C}_j| \geq 2M$, then all the clusters in group $H_t$ have size at least $M$. Now with probability at least $1 - \frac{2}{n^2}$, Membership$(v, \mathcal{C}_j) \geq$ Membership$(v, \mathcal{C}_{j''})$ for every cluster $\mathcal{C}_{j''} \in H_t$. In that case, the membership of $v$ is determined within at most $\lceil \log n \rceil$ queries. Else, with probability at most $\frac{2}{n^2}$, there may be $k$ queries to determine the membership of $v$.

Therefore, once a cluster has grown to size $2M$, the number of queries to resolve the membership of any vertex in those clusters is at most $\lceil \log n \rceil$ with probability at least $1 - \frac{2}{n^2}$. Hence, for at most $2kM$ elements, the number of queries made to resolve their membership can be $k$. Thus the total number of queries made by Algorithm 3 is $O(n \log n + Mk^2) = O(n \log n + \frac{k^2 \log n}{\mathcal{H}^2(f_+\|f_-)})$ with probability $1 - o_n(1)$. $\qquad\square$

**Remark 1.** *While for the more general setting with unknown $f_{i,j}s$ (distribution referring to similarity of cluster $i$ and $j$), we do not know how to extend this algorithm yet, if the parameters were known it is possible to extend our algorithm to such setting. We can calculate $M_i = O(\frac{\log n}{\min_{j:j \neq i} \mathcal{H}^2(f_{i,i}\|f_{i,j})})$, and thus whenever the $i$ th clusters grows to size $M_i$, remainder of its members can be inferred.*

Since we handle very generic distributions, our upper bounds are off by a factor of $O(\log n)$ from the lower bound. Tightening this bound, e.g. for sparse SBM to match the conjectured trade-off between queries and threshold remains an important open question.

## C  Zero Query and the Stochastic Block Model

Consider the case when we allow zero query to the oracle. The clustering has to be done just by using the side information matrix. This is a direct generalization to the well-known stochastic block model. Indeed, if $f_+$ is Bernoulli($p$) and $f_-$ is Bernoulli($q$), then the side information matrix is a binary matrix, as in the case of stochastic block model [1, 24, 11, 36].

It is clear that if the clustering input instance is adversarial, then it is impossible to recover the clusters with high probability. For example, think of the situation that $k - 1$ clusters are of size 1 each. In that case, one of these $k - 1$ small cluster points cannot be assigned to the correct cluster without querying, with a positive probability. Note that we will not be able to have such an argument later when querying is allowed, which makes that case significantly difficult.

Let us look at the scenario, when there are $k$ clusters of size $\frac{n}{k}$ each. Suppose $V = \sqcup_{i=1}^{k} V_i$ is the correct clustering. Consider the different clustering instances, that can be derived from the correct clustering, by swapping any two points $a \in V_i$ and $b \in V_j$, $i \neq j$. There are $K = \binom{k}{2}\frac{n^2}{k^2} = \frac{n^2}{2}(1 - 1/k)$ such different clusterings (partitions) possible. Let us consider these $K$ different cases as $K$ hypotheses, and try to identify which one of them is true based on the side information matrix.

Let $Q_t, t = 1, \ldots, K$ be the joint probability distributions of the side information matrix under hypothesis $t, t = 1, \ldots, K$. Also, let the correct clustering be the zeroth hypothesis and induces a joint probability distribution $Q_0$.

In this type of multi-hypothesis testing problem, a standard tool to lower bound probability of error is Fano's inequality. However, Fano's inequality in its usual form in hypothesis testing (see, [26, Thm. 7]) does not give the tightest possible result in our case. We instead use another form of Fano's inequality from [23, Thm. II.1 Eq. (5)] - therein taking $Q = Q_0$ and taking $f(x) = x \log x$, we have, the probability of error $P_e$ of this hypothesis testing problem (to identify between the $K$ hypotheses) given by,

$$\frac{1}{K}\sum_i D(Q_i\|Q_0) \geq (1 - P_e)\log(K(1 - P_e)) + P_e \log(KP_e/(K - 1))$$

where $D(f\|g)$ is the Kullback-Leibler (KL) divergence. The KL divergence between joint distribution of independent random variables is sum of the KL divergence of the marginals, and the only times when the distributions of $w_{i,j}$ differs under $Q_i$ and under $Q_0$ is when $i$ or $j$ belong to the two clusters where elements were swapped. There are $\frac{4n}{k}$ such instances, among them $\frac{2n}{k}$ contributes $D(f_+\|f_-)$ to the sum and $\frac{2n}{k}$ contributes $D(f_-\|f_+)$ to the sum. Therefore we obtain,

$$P_e \geq 1 - \frac{\frac{1}{K}\sum_i D(Q_i\|Q_0) + \log 2}{\log K} \geq 1 - \frac{\frac{2n}{k}\Delta(f_+\|f_1)}{\log \frac{n^2}{2}(1 - 1/k)} \approx 1 - \frac{n\Delta(f_+\|f_1)}{k \log n},$$

where $\Delta(f\|g) \equiv D(f\|g) + D(f\|g)$ .

One particular regime of interest in the literature of stochastic block model appear (see, [2, 36]) when, $f_+ \sim \text{Bernoulli}\left(\frac{a\log n}{n}\right)$ and $f_- \sim \text{Bernoulli}\left(\frac{b\log n}{n}\right)$. Then $D(f_+\|f_-) = \frac{a\log n}{n}\log\frac{a}{b} + \left(1 - \frac{a\log n}{n}\right)\log\frac{1 - \frac{a\log n}{n}}{1 - \frac{b\log n}{n}}$ and $\Delta(f_+\|f_-) = (a - b)\frac{\log n}{n}\left(\log\frac{a}{b} - \log\frac{1 - \frac{a\log n}{n}}{1 - \frac{b\log n}{n}}\right) \approx \frac{\log n}{n}\cdot(a - b)\log\frac{a}{b}$.
In this case, $P_e \geq 1 - \frac{a-b}{k}\log\frac{a}{b}$, and, $P_e > 0$ as long as $(a - b)\log\frac{a}{b} < k$. This lower bound can be improved by considering generalized versions of Fano's inequality involving Hellinger divergence.

In particular, by constructing a different hypothesis testing scenario and using a generalized version of Fano's inequality we can obtain the following result on probability $P_e$ of erroneous clustering. In particular, we can use a generalized version of Fano's inequality due to Polyanskiy and Vérdu [37, Thm. 4]. Consider the following different hypothesis testing situation. Suppose $k$ divides $n$, and there are $k$ equally sized subsets that partition the set of elements $[n] = V_1 \sqcup V_2 \sqcup \cdots \sqcup V_k$. Let $v \in V_1$ be a fixed element. Take any cluster $V_j, j \neq 1$. For all elements $u_1, \ldots, u_{n/k} \in V_j$, we obtain $K = n/k$ different hypotheses by interchanging $v$ with $u_i, i = 1, \ldots, n/k$. We consider the probability of error of this hypothesis testing problem. Inparticular, [37, Thm. 4], says that the probability of error $P_e$ is given by (considering Renyi divergence of order $\frac{1}{2}$),

$$-2\log\left(\sqrt{\frac{1-P_e}{K}} + \sqrt{P_e(1-\frac{1}{K})}\right) \leq -\log\sum_y (\frac{1}{K}\sum_{j=1}^K \sqrt{Q_j(y)})^2$$

which implies for us,

$$\left(\sqrt{\frac{1-P_e}{K}} + \sqrt{P_e(1-\frac{1}{K})}\right)^2 \geq \frac{1}{K^2}\sum_j\sum_i\sum_y \sqrt{Q_j(y)Q_i(y)} = \frac{1}{K^2}\sum_j\sum_i \left(1 - \mathcal{H}^2(Q_i\|Q_j)\right)$$

$$= 1 - \mathcal{H}^2(Q_i\|Q_j) = 1 - \left(1 - (1-\mathcal{H}^2(f_+\|f_-))^{\frac{4n}{k}}\right) = (1-\mathcal{H}^2(f_+\|f_-))^{\frac{4n}{k}},$$

where we had to crucially used the following fact: if $P_1^m$ and $Q_1^m$ denote joint distributions of $m$ of independent $P_i$ and independent $Q_i, i = 1, \ldots, m$ random variables, then,

$$\mathcal{H}^2(P_1^m\|Q_1^m) = 1 - \int_{x_1,\ldots,x_m} \sqrt{P_1^m(x_1,\ldots,x_m)Q_1^m(x_1,\ldots,x_m)}dx_1,\ldots,dx_m$$

$$= 1 - \prod_{i=1}^m \int_x \sqrt{P_i(x)Q_i(x)}dx \qquad\qquad \text{using Tonelli's theorem}$$

$$= 1 - \prod_{i=1}^m (1-\mathcal{H}^2(P_i\|Q_i)) \leq \sum_{i=1}^m \mathcal{H}^2(P_i\|Q_i).$$

Again, we assume $f_+ \sim \text{Bernoulli}\left(\frac{a\log n}{n}\right)$ and $f_- \sim \text{Bernoulli}\left(\frac{b\log n}{n}\right)$. In this case,

$$\sqrt{\frac{k}{n}} + \sqrt{P_e} \geq \left(\sqrt{ab}\frac{\log n}{n} + \sqrt{(1-\frac{a\log n}{n})(1-\frac{b\log n}{n})}\right)^{\frac{2n}{k}} = \left(1 - \left(\frac{a+b}{2} - \sqrt{ab} - \frac{ab\log n}{n}\right)\frac{\log n}{n}\right)^{\frac{2n}{k}}$$

$$\approx e^{-\left(\frac{a+b}{2} - \sqrt{ab} - \frac{ab\log n}{n}\right)\frac{2\log n}{k}} = n^{-\left(\frac{a+b}{2} - \sqrt{ab} - \frac{ab\log n}{n}\right)\frac{2}{k}}.$$

This implies, $\sqrt{P_e} \geq n^{-\left(\frac{a+b}{2} - \sqrt{ab}\right)\frac{2}{k}} - \sqrt{k}n^{-1/2}$ In particular, if $\left(\frac{a+b}{2} - \sqrt{ab}\right)\frac{2}{k} < \frac{1}{2}$, then $P_e > 0$. Hence, $P_e > 0$ if

$$\sqrt{a} - \sqrt{b} < \sqrt{\frac{k}{2}}.$$

While in this regime, this result is slightly suboptimal compared to the lower bound of [2], where the corresponding bound was $\sqrt{a} - \sqrt{b} < \sqrt{k}$, note that our bound works for arbitrary $f_+, f_-$ and across all regimes; moreover we have not tried to optimize the constants here.

## D Connections & Future Direction

This is the first work that rigorously study the query complexity of clustering with side information. We introduce new general information theoretic methods; as well as use, information theoretic inequalities to design efficient algorithms for clustering with near-optimal complexity. Our algorithms are entirely parameter free, and are computationally efficient. This work reveals interesting connection to the well-studied model of the stochastic block model and, generalize them in a significant way

by considering arbitrary distribution for noise opposed to only Bernoulli noise, and opens up new direction of study in the general area of clustering and community detection.

Even for the zero-query case, using generalized Fano's inequality in multiple hypothesis testing, we can derive simple lower bounds for SBM with arbitrary $f_+, f_-$ and cluster size distribution, matching closely the bounds for the sparse region $f_+ \sim \text{Bernoulli}(\frac{a \log n}{n})$ and $f_- \sim \text{Bernoulli}(\frac{b \log n}{n})$ and cluster size $\sim \frac{n}{k}$. Extending this lower bound to consider adaptive querying comes as a major challenge, as querying may reveal different deterministic information under different hypothesis. We propose a general framework for deriving such lower bounds, and in the process it reveals an interesting trend on how the threshold of recovery should change with querying: from $\sqrt{a} - \sqrt{b} \geq \sqrt{k}$ to $\sqrt{a} - \sqrt{b} \geq \sqrt{k}\left(1 - \frac{Q}{nk}\right)$ (see Lemma 1). That is querying can help reduce the threshold when $O(n)$ edges have been queried as $k$ is a constant. Currently, there is a $\sqrt{\log n}$ gap to achieve this bound as our lower bounds deal with very generic distributions and cluster sizes. Closing this gap for the stochastic block model with querying remains an interesting open question.

There is also a very recent result by [5] that studies the specific $k$-means clustering problem with a different side information model. While the setting is quite different, it is an interesting future work to improve their result (for example, they show a lower bound of $\Omega(\log k + \log n)$ to overcome NP-hardness of the problem) using our general methods.

## Footnotes

[4] the algorithm works even with different $f_{i,i}$s

[5]the precise value of $C$ can be deduced from the proof given $q$

[6]We could have also assumed the largest cluster size is at most $n(1-\epsilon)$ for some constant $\epsilon > 0$ and adjust the constants appropriately.

[7]note that the version of the Chernoff bound also holds for sampling without replacement, which is the case here [27].