[Reviews · NeurIPS 2017]

Reviewer 1



This paper studies query complexity for clustering with 'side information'. The 'side information' is provided as a similarity matrix W, whose Wij entry is distributed as f+ or f-, depending on if i,j belong to the same cluster or different cluster. The main result of the paper is that one requires Theta(nk) queries without side information whereas they show that only O(k log n/H^2(f+||f-) ) queries are sufficient with side information, where H is the Hellinger distance. They also show that up to a log(n) factor, this is tight. I find this to be an interesting theoretical result for several reasons. This is closely related to SBM which has seen a lot of activity in recent years, but asks some interesting questions as to how allowing queries change the threshold. It shows that allowing side information can drastically change query complexity. This itself is probably not surprising as the side information can have the complete cluster information, but the paper identifies how to assess the quality of the similarity matrix.

Reviewer 2



NOTE: I am reviewing two papers that appear to be by the same authors and on the same general topic. The other one is on noisy queries without any side information. This paper gives upper and lower bounds on the query complexity of clustering based on noiseless pairwise comparisons, along with random side-information associated with every pair. While I am familiar with some of the related literature, my knowledge of it is far from complete, so it's a little hard to fully judge the value of the contributions. The paper is also so dense that I couldn't spend as much time as ideal checking the correctness -- especially in the long upper bound proof. My overall impression of both papers is that the contributions seem to be valuable (if correct), but the writing could do with a lot of work. I acknowledge that the contributions are more important than the writing, but there are so many issues that my recommendation is still borderline. It would have been very useful to have another round of reviews where the suggestions are checked, but I understand this is not allowed for NIPS. Regarding this paper, there are a few very relevant works that were not cited: - The idea of replacing Bernoulli(p) and Bernoulli(q) by a general distribution in the stochastic block model is not new. A Google (or Google Scholar) search for "labeled [sometimes spelt labelled] stochastic block model" or the less common "weighted stochastic block model" reveals quite a few papers that have looked at similar generalizations. - There is also at least one work studying the SBM with an extra querying (active learning) element: "Active Learning for Community Detection in Stochastic Block Models" (Gadde et al, ISIT 2016). It is with a different querying model, where one only queries a single node and gets its label. But for fixed k (e.g., k=2 or k=100) and linear-size communities, this is basically the same as your pairwise querying model -- just quickly find a node from each community via random sampling, and then you can get the label of any new node by performing a pairwise query to each of those. At first glance it looks like your lower bound contradicts their (sublinear-sample) upper bound, but this might be because your lower bound only holds as k -> Infinity (see below) In any case, in light of these above works, I would ask the authors to remove/re-word all statements along the lines of "we initiate a rigorous theoretical study", "significantly generalizes them", "This is the first work that...", "generalize them in a significant way", "There is no systematic theoretical study", etc. Here are some further comments: - I think the model/problem should be presented more clearly. It looks like you are looking at minimax bounds, but this doesn't seem to be stated explicitly. Moreover, is the minimax model without any restriction on the cluster sizes? For instance, do you allow the case where there are k-1 clusters of size 1 and the other of size n-k+1? - It looks like Theorem 2 can't be correct as stated. For the SBM with k=2 (or probably any k=O(1)) we know that *no* queries are required even in certain cases where the Hellinger distance scales as log(n)/n. It looks like part if the problem is that the proof assumes k -> Infinity, which is not stated in the theorem (it absolutely needs to be). I'm not sure if that's the only part of the problem -- please check very carefully whether Theorem 2 more generally contradicts cases where clustering is possible even with Q=0. - The numerical section does not seem to be too valuable, mainly because it does not compare against any existing works, and only against very trivial baselines. Are your methods really the only ones that can be considered/implemented? - There are many confusing/unclear sentences, like "assume, otherwise the clustering is done" (is there text missing after "assume"?) and "the theorem is already proved from the nk lower bound" (doesn't that lower bound only apply when there's no side information?) - The paper is FULL of grammar mistakes, typos, and strange wording. Things like capitalization, plurals, spacing, and articles (a/an/the) are often wrong. Commas are consistently mis-used (e.g., they usually should not appear after statements like "let" / "we have" / "this means" / "assume that" / "since" / "while" / "note that", and very rarely need to be used just before an equation). Sentences shouldn't start with And/Or/Plus. The word "else" is used strangely (often "otherwise" would be better). Footnotes should start with a capital letter and end with a full stop. This list of issues is far from complete -- regardless of the decision, I strongly urge the authors to proof-read the paper with as much care as possible. Some less significant comments: - The abstract seems too long, and more generally, I don't think it's ideal to spend 3 pages before getting to the contributions - Is your use of terminology "Monte Carlo" and "Las Vegas" standard? Also, when you first mention Las Vegas on p4, it should be mentioned alongside *AVERAGE* query complexity (the word 'average' is missing) - Footnote 1: Should the 1- really be there after p= ? - In the formal problem statement "find Q subset of V x V" makes it sound like the algorithm is non-adaptive. - I disagree with the statement "clearly exhibiting" in the last sentence of the Simulations section. - Many brackets are too small throughout the appendix [POST-AUTHOR FEEDBACK COMMENTS] The authors clarified a few things in the responses, and I have updated the recommendation to acceptance. I still hope that a very careful revision is done for the final version considering the above comments. I can see that I was the only one of the 3 reviewers to have significant comments about the writing, but I am sure that some of these issues would make a difference to a lot more NIPS readers, particularly those coming from different academic backgrounds to the authors. Some specific comments: - The authors have the wrong idea if they feel that "issues like capitalization and misplaced commas" were deciding factors in the review. These were mentioned at the end as something that should be revised, not as a reason for the borderline recommendation. - Thank you for clarifying the (lack of) contradiction with Gadde's paper. - It is very important that theorems and lemmas are stated precisely. e.g., If a result only holds for sufficiently large k, this MUST be stated in the theorem, and not just in the proof. - Regarding the specific grammar issues addressed in the responses, I still suggest revising the use of Else/And/Or/Plus, the sentence "assume, otherwise...", and so on. I am convinced that most of these are grammatically incorrect as used in the paper, or at least unsuitable language for an academic paper. (The authors may disagree)